# Learning Low Dimensional State Spaces with Overparameterized Recurrent Neural Nets

**Edo Cohen-Karlik[1], Itamar Menuhin-Gruman[1], Raja Giryes[1], Nadav Cohen[1] & Amir Globerson[1,2]**
[1]Tel Aviv University [2]Google Research
{edocohen,menuhin}@mail.tau.ac.il
{raja,cohennadav,gamir}@tauex.tau.ac.il

## ABSTRACT

Overparameterization in deep learning typically refers to settings where a trained neural network (NN) has representational capacity to fit the training data in many ways, some of which generalize well, while others do not. In the case of Recurrent Neural Networks (RNNs), there exists an additional layer of overparameterization, in the sense that a model may exhibit many solutions that generalize well for sequence lengths seen in training, some of which *extrapolate* to longer sequences, while others do not. Numerous works have studied the tendency of Gradient Descent (GD) to fit overparameterized NNs with solutions that generalize well. On the other hand, its tendency to fit overparameterized RNNs with solutions that extrapolate has been discovered only recently and is far less understood. In this paper, we analyze the extrapolation properties of GD when applied to overparameterized linear RNNs. In contrast to recent arguments suggesting an implicit bias towards short-term memory, we provide theoretical evidence for learning low-dimensional state spaces, which can also model long-term memory. Our result relies on a dynamical characterization which shows that GD (with small step size and near-zero initialization) strives to maintain a certain form of balancedness, as well as on tools developed in the context of the *moment problem* from statistics (recovery of a probability distribution from its moments). Experiments corroborate our theory, demonstrating extrapolation via learning low-dimensional state spaces with both linear and non-linear RNNs.

## 1 INTRODUCTION

Neural Networks (NNs) are often *overparameterized*, in the sense that their representational capacity far exceeds what is necessary for fitting training data. Surprisingly, training overparameterized NNs via (variants of) Gradient Descent (GD) tends to produce solutions that generalize well, despite existence of many solutions that do not. This *implicit generalization* phenomenon attracted considerable scientific interest, resulting in various theoretical explanations (see, e.g., Woodworth et al. (2020); Yun et al. (2020); Zhang et al. (2017); Li et al. (2020); Ji & Telgarsky (2018); Lyu & Li (2019)).

Recent studies have surfaced a new form of implicit bias that arises in Recurrent Neural Networks (RNNs) and their variants (e.g., Long Short-Term Memory Hochreiter & Schmidhuber (1997) and Gated Recurrent Units Chung et al. (2014)). For such models, the length of sequences in training is often shorter than in testing, and it is not clear to what extent a learned solution will be able to *extrapolate* beyond the sequence lengths seen in training. In the overparameterized regime, where the representational capacity of the learned model exceeds what is necessary for fitting short sequences, there may exist solutions that generalize but do not extrapolate, meaning that their accuracy is high over short sequences but arbitrarily poor over long ones (see Cohen-Karlik et al. (2022)). In practice however, when training RNNs using GD, accurate extrapolation is often observed. We refer to this phenomenon as the *implicit extrapolation* of GD.

As opposed to the implicit generalization of GD, little is formally known about its implicit extrapolation. Existing theoretical analyses of the latter focus on linear RNNs — also known as *Linear Dynamical Systems* (LDS) — and either treat infinitely wide models (Emami et al., 2021), or models of finite width that learn from a memoryless teacher (Cohen-Karlik et al., 2022). In these regimes,

GD has been argued to exhibit an implicit bias towards short-term memory. While such results are informative, their generality remains in question, particularly since infinitely wide NNs are known to substantially differ from their finite-width counterparts, and since a memoryless teacher essentially neglects the main characteristic of RNNs (memory).

In this paper, we theoretically investigate the implicit extrapolation of GD when applied to overparameterized finite-width linear RNNs learning from a teacher with memory. We consider models with symmetric transition matrices, in the case where a student (learned model) with state space dimension $d$ is trained on sequences of length $k$ generated by a teacher with state space dimension $\hat{d}$. Our interest lies in the overparameterized regime, where $d$ is greater than both $k$ and $\hat{d}$, meaning that the student has state space dimensions large enough to fully agree with the teacher on sequences of length $k$, while potentially disagreeing with it on longer sequences. As a necessary assumption on initialization, we follow prior work and focus on a certain balancedness condition, which is known (see experiments in Cohen-Karlik et al. (2022), as well as our theoretical analysis) to capture near-zero initialization as commonly employed in practice.

Our main theoretical result states that GD originating from a balanced initialization leads the student to extrapolate, *irrespective of how large its state space dimension is*. Key to the result is a surprising connection to a moment matching theorem from Cohen & Yeredor (2011), whose proof relies on ideas from compressed sensing (Elad, 2010; Eldar & Kutyniok, 2012) and neighborly polytopes (Gale, 1963). This connection may be of independent interest, and in particular may prove useful in deriving other results concerning the implicit properties of GD. We corroborate our theory with experiments, which demonstrate extrapolation via learning low-dimensional state spaces in both the analyzed setting and ones involving non-linear RNNs.

The implicit extrapolation of GD is an emerging and exciting area of inquiry. Our results suggest that short-term memory is not enough for explaining it as previously believed. We hope the techniques developed in this paper will contribute to a further understanding of this phenomenon.

## 2 RELATED WORK

The study of linear RNNs, or LDS, has a rich history dating back to at least the early works of Kalman (Kalman, 1960; 1963). An extensively studied question relevant to extrapolation is that of *system identification*, which explores when the parameters of a teacher LDS can be recovered (see Ljung (1999)). Another related topic concerns finding compact realizations of systems, i.e. realizations of the same input-output mapping as a given LDS, with a state space dimension that is lower (see Antoulas (2005)). Despite the relation, our focus is fundamentally different from the above — we ask what happens when one learns an LDS using GD. Since GD is not explicitly designed to find a low-dimensional state space, it is not clear that the application of GD to an overparameterized student allows system identification through a compact realization. The fact that it does relate to the implicit properties of GD, and to our knowledge has not been investigated in the classic LDS literature.

The implicit generalization of GD in training RNNs has been a subject of theoretical study for at least several years (see, e.g., Hardt et al. (2016); Allen-Zhu et al. (2019); Lim et al. (2021)). In contrast, works analyzing the implicit extrapolation of GD have surfaced only recently, specifically in Emami et al. (2021) and Cohen-Karlik et al. (2022) Emami et al. (2021) analyzes linear RNNs in the infinite width regime, suggesting that in this case GD is implicitly biased towards impulse responses corresponding to short-term memory. Cohen-Karlik et al. (2022) studies finite-width linear RNNs (as we do), showing that when the teacher is memoryless (has state space dimension zero), GD emanating from a balanced initialization successfully extrapolates. Our work tackles an arguably more realistic and challenging setting — we analyze the regime in which the teacher has memory. Our results suggest that the implicit extrapolation of GD does not originate from a bias towards short-term memory, but rather a tendency to learn low-dimensional state spaces.[1] We note that there have been works studying extrapolation in the context of non-recurrent NNs, e.g. Xu et al. (2020).

---

[1] There is no formal contradiction between our results and those of (Emami et al., 2021) and (Cohen-Karlik et al., 2022). These works make restrictive assumptions (namely, the former assumes that the teacher is stable and its impulse response decays exponentially fast, and the latter assumes that the teacher is memoryless) under which implicit extrapolation via learning low-dimensional state spaces leads to solutions with short-term memory. Our work on the other hand is not limited by these assumptions, and we show that in cases where they are violated, learning yields solutions with low-dimensional state spaces that do *not* result in short-term memory.

This type of extrapolation deals with the behavior of learned functions outside the support of the training distribution, and thus fundamentally different from the type of extrapolation we consider, which deals with the behavior of learned functions over sequences longer than those seen in training.

Linear RNNs fundamentally differ from the commonly studied model of linear (feed-forward) NNs (see, e.g., Arora et al. (2018); Ji & Telgarsky (2018); Arora et al. (2019a;b); Razin & Cohen (2020)). One of the key differences is that a linear RNN entails different powers of a parameter (transition) matrix, leading to a loss function which roughly corresponds to a sum of losses for multiple linear NNs having different architectures and shared weights. This precludes the use of a vast array of theoretical tools tailored for linear NNs, rendering the analysis of linear RNNs technically challenging.

On the empirical side, extrapolation of NNs to sequence lengths beyond those seen in training has been experimentally demonstrated in numerous recent works, covering both modern language and attention models (Press et al., 2022; Anil et al., 2022; Zhang et al., 2022), and RNNs with transition matrices of particular forms (Gu et al., 2022; 2021; 2020; Gupta, 2022). The current paper is motivated by these findings, and takes a step towards theoretically explaining them.

## 3 LINEAR RECURRENT NEURAL NETWORKS

Our theoretical analysis applies to single-input single-output (SISO) linear RNNs with symmetric transition matrices. Given a state space dimension $d \in \mathbb{N}$, this model is defined by the update rule:

$$\boldsymbol{s}_{t+1} = \boldsymbol{A}\boldsymbol{s}_t + \boldsymbol{B}x_t, \quad y_t = \boldsymbol{C}\boldsymbol{s}_t, \quad t = 0, 1, 2, \ldots, \tag{3.1}$$

where $\boldsymbol{A} \in \mathbb{R}^{d \times d}$, $\boldsymbol{B} \in \mathbb{R}^{d \times 1}$ and $\boldsymbol{C} \in \mathbb{R}^{1 \times d}$ are configurable parameters, the *transition matrix* $\boldsymbol{A}$ satisfies $\boldsymbol{A} = \boldsymbol{A}^\top$; $x_0, x_1, \ldots \in \mathbb{R}$ form an input sequence; $y_0, y_1, \ldots \in \mathbb{R}$ form the corresponding output sequence; and $\boldsymbol{s}_t \in \mathbb{R}^{d \times 1}$ represents the internal state at time $t$, assumed to be equal to zero at the outset (i.e. it is assumed that $\boldsymbol{s}_0 = \boldsymbol{0}$). As with any linear time-invariant system (Porat, 1996), the input-output mapping realized by the RNN is determined by its impulse response.

**Definition 1.** *The **impulse response** of the RNN is the output sequence corresponding to the input sequence $(x_0, x_1, x_2, \ldots) = (1, 0, 0, \ldots)$. Namely, it is the sequence $(\boldsymbol{CB}, \boldsymbol{CAB}, \boldsymbol{CA}^2\boldsymbol{B}, \ldots)$.*

For brevity, we employ the shorthand $\Theta := (\boldsymbol{A}, \boldsymbol{B}, \boldsymbol{C})$. The $d \times d$ symmetric transition matrix $\boldsymbol{A}$ is parameterized through a $d(d+1)/2$-dimensional vector holding its upper triangular elements, and with a slight overloading of notation, the symbol $\boldsymbol{A}$ is also used to refer to this parameterization.

We note that our theory readily extends to multiple-input multiple-output (MIMO) networks, and the focus on the SISO case is merely for simplicity of presentation. Note also that the restriction to symmetric transition matrices is customary in both theory (Hazan et al., 2018) and practice (Gupta, 2022), and represents a generalization of the *canonical modal form*, which under mild non-degeneracy conditions does not limit generality (Boyd & Lessard, 2006).

Given a length $k$ input sequence, $\mathbf{x} = (x_0, \ldots, x_{k-1}) \in \mathbb{R}^k$, consider the output at the last time step, i.e. $y := y_k \in \mathbb{R}$, and denote it by $RNN(\boldsymbol{x})$. Using this output as a label, we define an empirical loss induced by a training set $S = \left\{ \left( \boldsymbol{x}^{(1)}, y^{(1)} \right), \ldots, \left( \boldsymbol{x}^{(N)}, y^{(N)} \right) \right\} \subset \mathbb{R}^k \times \mathbb{R}$:

$$\mathcal{L}_S(\boldsymbol{A}, \boldsymbol{B}, \boldsymbol{C}) = \frac{1}{N} \sum_{i=1}^N \ell \left( RNN \left( \boldsymbol{x}^{(i)} \right), y^{(i)} \right), \tag{3.2}$$

where $\ell(y, \hat{y}) = (y - \hat{y})^2$ is the square loss. By the update rule of the RNN (Equation 3.1), we have:

$$\mathcal{L}_S(\boldsymbol{A}, \boldsymbol{B}, \boldsymbol{C}) = \frac{1}{N} \sum_{i=1}^N \left( \sum_{j=0}^{k-1} \boldsymbol{CA}^{k-1-j} \boldsymbol{B} x_j^{(i)} - y^{(i)} \right)^2. \tag{3.3}$$

Suppose that ground truth labels are generated by an RNN as defined in Equation 3.1, and denote the state space dimension and parameters of this *teacher* network by $\hat{d}$ and $\hat{\Theta} = (\hat{\boldsymbol{A}}, \hat{\boldsymbol{B}}, \hat{\boldsymbol{C}})$ respectively. We employ the common assumption (e.g., see Hardt et al. (2016)) by which input sequences are drawn from a whitened distribution, i.e. a distribution where $\mathbb{E}[x_j x_{j'}]$ equals 1 if $j = j'$ and 0 otherwise. The population loss over length $k$ sequences can then be written as (see Lemma E.1):

$$\mathcal{L}(\boldsymbol{A}, \boldsymbol{B}, \boldsymbol{C}) = \sum_{j=0}^{k-1} \left( \boldsymbol{CA}^j \boldsymbol{B} - \hat{\boldsymbol{C}} \hat{\boldsymbol{A}}^j \hat{\boldsymbol{B}} \right)^2. \tag{3.4}$$

Equation 3.4 implies that a solution $\Theta = (\boldsymbol{A}, \boldsymbol{B}, \boldsymbol{C})$ achieves zero population loss over length $k$ sequences if and only if $\boldsymbol{C}\boldsymbol{A}^j\boldsymbol{B} = \hat{\boldsymbol{C}}\hat{\boldsymbol{A}}^j\hat{\boldsymbol{B}}$ for $j = 0, \ldots, k-1$. To what extent does such a solution imply that the *student* (i.e., the learned RNN) extrapolates to longer sequences? This depends on how close $\boldsymbol{C}\boldsymbol{A}^j\boldsymbol{B}$ is to $\hat{\boldsymbol{C}}\hat{\boldsymbol{A}}^j\hat{\boldsymbol{B}}$ for $j \geqslant k$.

**Definition 2.** *For $\epsilon \geqslant 0$ and $q \in \mathbb{N}$, we say that the student $\epsilon$-**extrapolates with horizon** $q$ with respect to (w.r.t) the teacher if:*

$$|\boldsymbol{C}\boldsymbol{A}^j\boldsymbol{B} - \hat{\boldsymbol{C}}\hat{\boldsymbol{A}}^j\hat{\boldsymbol{B}}| \leqslant \epsilon, \quad \forall j \in \{0, 1, \ldots, q-1\}. \tag{3.5}$$

*If the above holds for all $q \in \mathbb{N}$ then the student is said to $\epsilon$-**extrapolate** w.r.t the teacher, and if it holds for all $q \in \mathbb{N}$ with $\epsilon = 0$ then the student is simply said to **extrapolate** w.r.t the teacher.*

Per Definition 2, $\epsilon$-extrapolation with horizon $q$ is equivalent to the first $q$ elements of the student's impulse response being $\epsilon$-close to those of the teacher's, whereas extrapolation means that the student's impulse response fully coincides with the teacher's. The latter condition implies that the student realizes the same input-output mapping as the teacher, for any sequence length (this corresponds to the notion of system identification; see Section 2).

Notice that when the student is *overparameterized*, in the sense that $d$ is greater than $k$ and $\hat{d}$, it may perfectly generalize, i.e. lead the population loss over length $k$ sequences (Equation 3.4) to equal zero, and yet fail to extrapolate, as stated in the following proposition.

**Proposition 3.** *Assume $d > k$, and let $\epsilon \geqslant 0$ and $q \in \{k+1, k+2, \ldots\}$. Then, for any teacher parameters $\hat{\Theta}$, there exist student parameters $\Theta$ with which the population loss in Equation 3.4 equals zero, and yet the student does **not** $\epsilon$-extrapolate with horizon $q$.*

*Proof sketch (for complete proof see Appendix E.1.2).* The result follows from the fact that the first $d$ elements of the student's impulse response can be assigned freely via a proper choice of $\Theta$. $\square$

We are interested in the extent to which student parameters learned by GD extrapolate in the overparameterized regime. Proposition 3 implies that, regardless of how many (length $k$) sequences are used in training, if GD leads to any form of extrapolation, it must be a result of some implicit bias induced by the algorithm. Note that in our setting, extrapolation cannot be explained via classic tools from statistical learning theory, as evaluation over sequences longer than those seen in training violates the standard assumption of train and test data originating from the same distribution.

To decouple the question of extrapolation from that of generalization, we consider the case where the training set $S$ is large, or more formally, where the empirical loss $\mathcal{L}_S(\cdot)$ (Equation 3.3) is well represented by the population loss $\mathcal{L}(\cdot)$ (Equation 3.4). We model GD with small step size via Gradient Flow (GF), as customary in the theory of NNs — see Saxe et al. (2013); Gunasekar et al. (2017); Arora et al. (2018; 2019b); Lyu & Li (2019); Li et al. (2020); Azulay et al. (2021) for examples where it is used and Elkabetz & Cohen (2021) for a theoretical justification of its usage. Using the GF formulation, we analyze the following dynamics:

$$\dot{\alpha}(\tau) := \frac{d}{d\tau}\alpha(\tau) = -\frac{\partial}{\partial\alpha}\mathcal{L}\big(\boldsymbol{A}(\tau), \boldsymbol{B}(\tau), \boldsymbol{C}(\tau)\big), \ \tau \geqslant 0, \tag{3.6}$$

where $\alpha \in \{\boldsymbol{A}, \boldsymbol{B}, \boldsymbol{C}\}$. If no assumption on initialization is made, no form of extrapolation can be established (indeed, the initial point may be a global minimizer of $\mathcal{L}(\cdot)$ that fails to extrapolate, and GF will stay there). Following prior work (see Cohen-Karlik et al. (2022)), we assume that the initialization adheres to the following balancedness condition:

**Definition 4.** *An RNN with parameters $\Theta = (\boldsymbol{A}, \boldsymbol{B}, \boldsymbol{C})$ is said to be **balanced** if $\boldsymbol{B} = \boldsymbol{C}^\top$.*

It was shown empirically in Cohen-Karlik et al. (2022) that the balancedness condition captures near-zero initialization as commonly employed in practice. We support this finding theoretically in Section 4.3. Aside from the initialization of the student, we will also assume that the teacher adheres to the balancedness condition.

## 4 THEORETICAL ANALYSIS

We turn to our theoretical analysis. Section 4.1 proves that in the setting of Section 3, convergence of GF to a zero loss solution leads the student to extrapolate, *irrespective of how large its state space*

*dimension is*. Section 4.2 extends this result by establishing that, under mild conditions, approximate convergence leads to approximate extrapolation. The results of sections 4.1 and 4.2 assume that GF emanates from a balanced initialization, which empirically is known to capture near-zero initialization as commonly employed in practice (see Section 3). Section 4.3 theoretically supports this empirical premise, by showing that with high probability, random near-zero initialization leads to balancedness.

We introduce notations that will be used throughout the analysis. For a matrix $\boldsymbol{Q} \in \mathbb{R}^{m \times n}$, we let $\|\boldsymbol{Q}\|_F$, $\|\boldsymbol{Q}\|_\infty$ and $\|\boldsymbol{Q}\|_2$ denote the Frobenius, $\ell_\infty$ and $\ell_2$ (spectral) norms, respectively. For a vector $\boldsymbol{v} \in \mathbb{R}^m$, we use $\|\boldsymbol{v}\|$ to denote the Euclidean norm and $v_i$ to denote its $i^{th}$ entry.

## 4.1 CONVERGENCE LEADS TO EXTRAPOLATION

Theoretical analyses of implicit generalization often assume convergence to a solution attaining zero loss (see, e.g., Azulay et al. (2021); Gunasekar et al. (2017); Lyu & Li (2019); Woodworth et al. (2020)). Under such an assumption, Theorem 5 below establishes implicit extrapolation, i.e. that the solution to which GF converges extrapolates, *irrespective of how large the student's state space dimension $d$ is*. A condition posed by the theorem is that the training sequence length $k$ is greater than two times the teacher's state space dimension $\hat{d}$. This condition is necessary — see Appendix A for theoretical justification and Section 5 for empirical demonstration.

**Theorem 5.** *Assume that $d > k > 2\hat{d}$, the teacher parameters $\hat{\Theta}$ are balanced (Definition 4), and the student parameters $\Theta$ are learned by applying GF to the loss $\mathcal{L}(\cdot)$ (Equation 3.4) starting from a balanced initialization. Then, if GF converges to a point $\Theta^*$ satisfying $\mathcal{L}(\Theta^*) = 0$, this point extrapolates (Definition 2).*

In order to prove Theorem 5, we introduce two lemmas: Lemma 6, which shows that balancedness is preserved under GF; and Lemma 7, which (through a surprising connection to a moment problem from statistics) establishes that a balanced solution attaining zero loss necessarily extrapolates. With Lemmas 6 and 7 in place, the proof of Theorem 5 readily follows.

**Lemma 6.** *Let $\Theta(\tau)$, with $\tau \geqslant 0$, be a curve brought forth by applying GF to the loss $\mathcal{L}(\cdot)$ starting from a balanced initialization. Then, $\Theta(\tau)$ is balanced for every $\tau \geqslant 0$.*

*Proof sketch (for complete proof see Appendix E.1.5).* The result follows from the symmetric role of $\boldsymbol{B}$ and $\boldsymbol{C}$ in the loss $\mathcal{L}(\cdot)$. □

**Lemma 7.** *Suppose that $d > k > 2\hat{d}$, the teacher is balanced, and that the student parameters $\Theta$ are balanced and satisfy $\mathcal{L}(\Theta) = 0$. Then $\Theta$ extrapolates.*

*Proof sketch (for complete proof see Appendix E.2).* The proof is based on a surprising connection that we draw to the *moment problem* from statistics (recovery of a probability distribution from its moments), which has been studied for decades (see, e.g., Schmüdgen (2017)).

Without loss of generality, we may assume that $\hat{\boldsymbol{A}}$ is diagonal (if this is not the case then we apply an orthogonal eigendecomposition to $\hat{\boldsymbol{A}}$ and subsequently absorb eigenvectors into $\hat{\boldsymbol{B}}$ and $\hat{\boldsymbol{C}}$). We may also assume that $\hat{\boldsymbol{C}}\hat{\boldsymbol{B}} = 1$ (otherwise we absorb a scaling factor into $\boldsymbol{B}$ and/or $\boldsymbol{C}$). Since $\hat{\boldsymbol{C}}^\top = \hat{\boldsymbol{B}}$ (teacher is balanced), we may define a probability vector (i.e. a vector with non-negative entries summing up to one) $\hat{\boldsymbol{p}} \in \mathbb{R}^{\hat{d}}$ via $\hat{p}_i = \hat{C}_i\hat{B}_i$, $i = 1, \ldots, \hat{d}$. We let $\hat{Z}$ denote the random variable supported on $\{\hat{A}_{1,1}, \ldots, \hat{A}_{\hat{d},\hat{d}}\}$, which assumes the value $\hat{A}_{i,i}$ with probability $\hat{p}_i$, $i = 1, \ldots, \hat{d}$. Notice that for every $j \in \mathbb{N}$:

$$\hat{\boldsymbol{C}}\hat{\boldsymbol{A}}^j\hat{\boldsymbol{B}} = \sum\nolimits_{i=1}^{\hat{d}} \hat{p}_i \hat{A}_{i,i}^j = \mathbb{E}[\hat{Z}^j],$$

meaning that the elements of the teacher's impulse response are precisely the moments of $\hat{Z}$.

Similarly to above we may assume $\boldsymbol{A}$ is diagonal, and since $\mathcal{L}(\Theta) = 0$ it holds that $\boldsymbol{C}\boldsymbol{B} = \hat{\boldsymbol{C}}\hat{\boldsymbol{B}} = 1$. We may thus define a probability vector $\boldsymbol{p} \in \mathbb{R}^d$ via $p_i = C_iB_i$, $i = 1, \ldots, d$, and a random variable $Z$ which assumes the value $A_{ii}$ with probability $p_i$, $i = 1, \ldots, d$. For every $j \in \mathbb{N}$:

$$\boldsymbol{C}\boldsymbol{A}^j\boldsymbol{B} = \sum\nolimits_{i=1}^{d} p_i A_{i,i}^j = \mathbb{E}[Z^j],$$

and so the elements of the student's impulse response are precisely the moments of $Z$.

The probabilistic formulation we set forth admits an interpretation of extrapolation as a moment problem. Namely, since $\mathcal{L}(\Theta) = 0$ (i.e. $\boldsymbol{C}\boldsymbol{A}^j\boldsymbol{B} = \hat{\boldsymbol{C}}\hat{\boldsymbol{A}}^j\hat{\boldsymbol{B}}$ for $j = 0, \ldots, k-1$) the random variables $Z$ and $\hat{Z}$ agree on their first $k-1$ moments, and the question is whether they agree on all higher moments as well. We note that this question is somewhat more challenging than that tackled in classic instances of the moment problem, since the support of the random variable whose moments we match ($Z$) is not known to coincide with the support of the random variable we seek to recover ($\hat{Z}$). Luckily, a recent powerful result allows addressing the question we face — Cohen & Yeredor (2011) showed that the first $2n$ moments of a discrete random variable $X$ taking at most $n \in \mathbb{N}$ values uniquely define $X$, in the sense that *any* discrete random variable agreeing with these $2n$ moments must be identical to $X$. Translating this result to our setting, we have that if $Z$ agrees with $\hat{Z}$ on its first $2\hat{d}$ moments, it must be identical to $\hat{Z}$, and in particular it must agree with $\hat{Z}$ on all higher moments as well. The fact that $k - 1 \geqslant 2\hat{d}$ then concludes the proof.

To attain some intuition for the result we imported from Cohen & Yeredor (2011), consider the simple case where $\hat{d} = 1$. The transition matrix $\hat{\boldsymbol{A}}$ is then a scalar $\hat{a} \in \mathbb{R}$, the random variable $\hat{Z}$ is deterministically equal to $\hat{a}$, and the teacher's impulse response is given by the moments $\mathbb{E}[\hat{Z}^j] = \hat{a}^j$, $j = 0, 1, \ldots$. Since we assume $k > 2\hat{d}$, the fact that $\mathcal{L}(\Theta) = 0$ means the random variable corresponding to the student, $Z$, agrees with the first two moments of $\hat{Z}$. That is, $Z$ satisfies $\mathbb{E}[Z] = \hat{a}$ and $\mathbb{E}[Z^2] = \hat{a}^2$. This implies that $\mathrm{Var}[Z] = \mathbb{E}[Z^2] - \mathbb{E}[Z]^2 = 0$, and therefore $Z$ is deterministically equal to $\hat{a}$, i.e. it is identical to $\hat{Z}$. The two random variables thus agree on all of their moments, meaning the impulse responses of the student and teacher are the same. $\square$

*Proof of Theorem 5.* By Lemma 6 (as well as continuity considerations) $\Theta^*$ is balanced. Therefore, Lemma 7 implies that it extrapolates. $\square$

## 4.2 Approximate Convergence Leads to Approximate Extrapolation

Theorem 5 in Section 4.1 proves extrapolation in the case where GF converges to a zero loss solution. Theorem 8 below extends this result by establishing that, under mild conditions, approximate convergence leads to approximate extrapolation — or more formally — for any $\epsilon > 0$ and $q \in \mathbb{N}$, when GF leads the loss to be sufficiently small, the student $\epsilon$-extrapolates with horizon $q$.

**Theorem 8.** *Assume the conditions of Theorem 5, and that the teacher parameters $\hat{\Theta}$ are stable, i.e. the eigenvalues of $\hat{\boldsymbol{A}}$ are in $[-1, 1]$. Assume also that $\hat{\Theta}$ are non-degenerate, in the sense that the input-output mapping they realize is not identically zero. Finally, assume that the student parameters $\Theta$ learned by GF are confined to some bounded domain in parameter space. Then, for any $\epsilon > 0$ and $q \in \mathbb{N}$, there exists $\delta(\epsilon, q) > 0$ such that whenever $\mathcal{L}(\Theta) \leqslant \delta(\epsilon, q)$, the student $\epsilon$-extrapolates with horizon $q$.*

*Proof sketch (for complete proof see Appendix E.3).* Let $\delta > 0$ be a constant whose value will be chosen later, and suppose GF reached a point $\Theta$ satisfying $\mathcal{L}(\Theta) \leqslant \delta$.

Following the proof of Lemma 7, $\hat{\Theta}$ is identified with a distribution supported on the eigenvalues of $\hat{\boldsymbol{A}}$, whose $j$th moment is $\hat{m}_j := \hat{\boldsymbol{C}}\hat{\boldsymbol{A}}^j\hat{\boldsymbol{B}}(\hat{\boldsymbol{C}}\hat{\boldsymbol{B}})^{-1}$ for every $j \in \mathbb{N}$. Similarly, $\Theta$ is identified with a distribution supported on the eigenvalues of $\boldsymbol{A}$, whose $j$th moment is $m_j := \boldsymbol{C}\boldsymbol{A}^j\boldsymbol{B}(\boldsymbol{C}\boldsymbol{B})^{-1}$ for every $j \in \mathbb{N}$. The fact that $\mathcal{L}(\Theta) \leqslant \delta$ implies $|\boldsymbol{C}\boldsymbol{B} - \hat{\boldsymbol{C}}\hat{\boldsymbol{B}}| \leqslant \sqrt{\delta}$, and in addition $|\hat{m}_j - m_j| \leqslant \mathcal{O}(\sqrt{\delta})$ for every $j = 1, \ldots, k-1$. To conclude the proof it suffices to show that

$$|\hat{m}_j - m_j| \leqslant \mathcal{O}(\epsilon) \quad \forall j \in \{1, \ldots, q-1\} \tag{4.1}$$

given a small enough choice for $\delta$ (this choice then serves as $\delta(\epsilon, q)$ in the theorem statement).

We establish Equation 4.1 by employing the theory of Wasserstein distances (Vaserstein, 1969). For $p \in \mathbb{N}$, denote by $\mathcal{W}_p$ the $p$-Wasserstein distance between the distributions identified with $\hat{\Theta}$ and $\Theta$. Since $k > 2\hat{d}$, it holds that $|\hat{m}_j - m_j| \leqslant \mathcal{O}(\sqrt{\delta})$ for every $j = 1, \ldots, 2\hat{d}$. Proposition 2 in Wu & Yang (2020) then implies $\mathcal{W}_1 \leqslant \mathcal{O}(\delta^{1/4\hat{d}})$. For any $p \in \mathbb{N}$, $\mathcal{W}_p \leqslant \mathcal{O}(\mathcal{W}_1^{1/p})$ (see Section 2.3 in Panaretos & Zemel (2019)) and $|\hat{m}_p - m_p| \leqslant \mathcal{O}(\mathcal{W}_p)$ (see Section 1.2 in Biswas & Mackey (2021)). Combining the latter three inequalities, we have that $|\hat{m}_p - m_p| \leqslant \mathcal{O}(\delta^{1/4\hat{d}p})$ for any $p \in \mathbb{N}$. Choosing $\delta = \mathcal{O}(\epsilon^{4\hat{d}(q-1)})$ therefore establishes Equation 4.1. $\square$

### 4.3 BALANCEDNESS CAPTURES NEAR-ZERO INITIALIZATION

Theorems 5 and 8 assume that GF emanates from a balanced initialization, i.e. from a point $\Theta = (\boldsymbol{A}, \boldsymbol{B}, \boldsymbol{C})$ satisfying $\boldsymbol{B} = \boldsymbol{C}^{\top}$. It was shown in Cohen-Karlik et al. (2022) that theoretical predictions derived assuming balanced initialization faithfully match experiments conducted with near-zero initialization (an initialization commonly used in practice). Proposition 9 below theoretically supports this finding, establishing that with high probability, random near-zero initialization leads GF to arrive at an approximately balanced point, i.e. a point $\Theta = (\boldsymbol{A}, \boldsymbol{B}, \boldsymbol{C})$ for which the difference between $\boldsymbol{B}$ and $\boldsymbol{C}^{\top}$ is negligible compared to their size.

**Proposition 9.** *Suppose that:* (i) $d > 20$; (ii) *the teacher parameters $\hat{\Theta}$ are balanced and are non-degenerate, in the sense that the input-output mapping they realize is not identically zero; and* (iii) *the student parameters are learned by applying GF to the loss $\mathcal{L}(\cdot)$. Let $\tilde{\Theta}$ be a random point in parameter space, with entries drawn independently from the standard normal distribution. For $\epsilon > 0$, consider the case where GF emanates from the initialization $\epsilon\tilde{\Theta}$, and denote the resulting curve by $\Theta_{\epsilon}(\tau) = (\boldsymbol{A}_{\epsilon}(\tau), \boldsymbol{B}_{\epsilon}(\tau), \boldsymbol{C}_{\epsilon}(\tau))$, with $\tau \geqslant 0$. Then, w.p. at least 0.75, for every $\epsilon > 0$ there exists $\tau_{\epsilon} \geqslant 0$ such that:*

$$\lim_{\epsilon \to 0^+} \frac{||\boldsymbol{B}_{\epsilon}(\tau_{\epsilon}) - \boldsymbol{C}_{\epsilon}^{\top}(\tau_{\epsilon})||_F}{||\boldsymbol{B}_{\epsilon}(\tau_{\epsilon}) + \boldsymbol{C}_{\epsilon}^{\top}(\tau_{\epsilon})||_F} = 0. \tag{4.2}$$

*Proof sketch (for complete proof see Appendix E.4).* The idea behind the proof is as follows. Assume $\epsilon$ is sufficiently small. Then, when the entries of $\Theta = (\boldsymbol{A}, \boldsymbol{B}, \boldsymbol{C})$ are on the order of $\epsilon$, we have $\frac{\partial}{\partial \boldsymbol{B}}\mathcal{L}(\Theta) \approx -2\hat{\boldsymbol{C}}\hat{\boldsymbol{B}} \cdot \boldsymbol{C}^{\top}$ and $\frac{\partial}{\partial \boldsymbol{C}}\mathcal{L}(\Theta) \approx -2\hat{\boldsymbol{C}}\hat{\boldsymbol{B}} \cdot \boldsymbol{B}^{\top}$. This implies that during the first part of the curve $\Theta_{\epsilon}(\tau)$ it holds that $\frac{d}{d\tau}(\boldsymbol{B}_{\epsilon}(\tau) - \boldsymbol{C}_{\epsilon}^{\top}(\tau)) \approx -2\hat{\boldsymbol{C}}\hat{\boldsymbol{B}} \cdot (\boldsymbol{B}_{\epsilon}(\tau) - \boldsymbol{C}_{\epsilon}^{\top}(\tau))$ and similarly $\frac{d}{d\tau}(\boldsymbol{B}_{\epsilon}(\tau) + \boldsymbol{C}_{\epsilon}^{\top}(\tau)) \approx 2\hat{\boldsymbol{C}}\hat{\boldsymbol{B}} \cdot (\boldsymbol{B}_{\epsilon}(\tau) + \boldsymbol{C}_{\epsilon}^{\top}(\tau))$. Since $\hat{\boldsymbol{C}}\hat{\boldsymbol{B}} > 0$ (follows from the teacher parameters being balanced and non-degenerate), the entries of $\boldsymbol{B}_{\epsilon}(\tau) - \boldsymbol{C}_{\epsilon}^{\top}(\tau)$ shrink exponentially fast while those of $\boldsymbol{B}_{\epsilon}(\tau) + \boldsymbol{C}_{\epsilon}^{\top}(\tau)$ grow at the same rate. This exponential shrinkage/growth leads $\|\boldsymbol{B}_{\epsilon}(\tau) - \boldsymbol{C}_{\epsilon}^{\top}(\tau)\|/\|\boldsymbol{B}_{\epsilon}(\tau) + \boldsymbol{C}_{\epsilon}^{\top}(\tau)\|$ to become extremely small, more so the smaller $\epsilon$ is. $\square$

## 5 EXPERIMENTS

In this section we present experiments corroborating our theoretical analysis (Section 4). The latter establishes that, under certain conditions, a linear RNN with state space dimension $d$ extrapolates when learning from a teacher network with state space dimension $\hat{d}$ via training sequences of length $k$, irrespective of how large $d$ is compared to $\hat{d}$ and $k$. A key condition underlying the result is that $k$ is larger than $2\hat{d}$. Section 5.1 below considers the theoretically analyzed setting, and empirically evaluates extrapolation as $k$ varies. Its results demonstrate a phase transition, in the sense that extrapolation takes place when $k > 2\hat{d}$, in compliance with theory, but fails when $k$ falls below $2\hat{d}$, in which case the theory indeed does not guarantee extrapolation. Section 5.2 displays the same phenomenon with linear RNNs that do not adhere to some of the assumptions made by the theory (in particular the assumption of symmetric transition matrices, and those concerning balancedness). Finally, Section 5.3 considers non-linear RNNs (specifically, Gated Recurrent Unit networks Chung et al. (2014)), and shows that they too exhibit a phase transition in extrapolation as the training sequence length varies. For brevity, we defer some of the details behind our implementation, as well as additional experiments, to Appendix B.

### 5.1 THEORETICALLY ANALYZED SETTING

Our first experiment considers the setting described in Section 3 and theoretically analyzed in Section 4. As representative values for the state space dimensions of the teacher and (overparameterized) student, we choose $\hat{d} = 5$ and $d = 40$ respectively (higher state space dimensions for the student, namely $d = 100$ and $d = 200$, yield qualitatively identical results). For a given training sequence length $k$, the student is learned via GD applied directly to the population loss defined in Equation 3.4 (applying GD to the empirical loss defined in Equation 3.3, with $N = 10,000$ training examples, led to similar results). Figure 1(a) reports the extrapolation error (quantified by the $\ell_{\infty}$ distance between the impulse response of the learned student and that of the teacher) as a function of $k$. As can be seen, extrapolation exhibits a phase transition that accords with our theory: when $k > 2\hat{d}$ extrapolation error is low, whereas when $k$ falls below $2\hat{d}$ extrapolation error is high.

## 5.2   OTHER SETTINGS WITH LINEAR RECURRENT NEURAL NETWORKS

To assess the generality of our findings, we experiment with linear RNNs in settings that do not adhere to some of the assumptions made by our theory. Specifically, we evaluate settings in which: *(i)* the teacher is unbalanced, meaning $\hat{B} \neq \hat{C}^\top$, and its transition matrix $\hat{A}$ is non-symmetric; *(ii)* the student's transition matrix $A$ is not restricted to be symmetric; *(iii)* learning is implemented by optimizing the empirical loss defined in Equation 3.3 (rather than the population loss defined in Equation 3.4); and *(iv)* optimization is based on Adam Kingma & Ba (2014) (rather than GD), emanating from standard near-zero initialization which is generally unbalanced (namely, $B \neq C^\top$). Figure 1(b) reports the results of an experiment where the state space dimensions of the teacher and (overparameterized) student are $\hat{d} = 10$ and $d = 50$ respectively (higher state space dimensions for the student, namely $d = 100$ and $d = 200$, yield qualitatively identical results), and where the teacher implements a delay line of $\hat{d}$ time steps (for details see Appendix C.2.2). Similar results obtained with randomly generated teachers are reported in Appendix B. As can be seen, despite the fact that our theory does not apply to the evaluated settings, its conclusions still hold — extrapolation error is low when the training sequence length $k$ is greater than $2\hat{d}$, and high when $k$ falls below $2\hat{d}$.

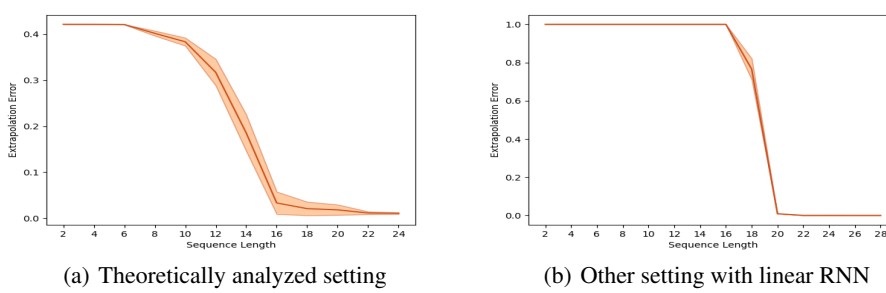

(a) Theoretically analyzed setting          (b) Other setting with linear RNN

Figure 1: Demonstration of implicit extrapolation with linear RNNs. Plots show extrapolation error (average over three random seeds, with shaded region marking standard deviation) as a function of training sequence length $k$, for a student with state space dimension $d$ learning from a teacher with state space dimension $\hat{d}$, where $d \gg \hat{d}$. (a) Models adhere to the setting described in Section 3 and theoretically analyzed in Section 4, with $\hat{d} = 5$, $d = 40$. (b) Models do not adhere to some of the assumptions made by the theory, and $\hat{d} = 10$, $d = 50$. Notice that extrapolation exhibits a phase transition that accords with theory — when $k > 2\hat{d}$ extrapolation error is low, and when $k$ falls below $2\hat{d}$ extrapolation error is high. The gradual transition exhibited is due to numerical errors introduced by the optimization not reaching an exact global minimum. For further details see Sections 5.1 and 5.2 and Appendix B.

## 5.3   NON-LINEAR RECURRENT NEURAL NETWORKS

As a final experiment, we explore implicit extrapolation with non-linear RNNs, namely GRU networks. Specifically, we evaluate the extent to which a student GRU with state space dimension $d_g = 100$ extrapolates when learning from a teacher GRU with state space dimension $\hat{d}_g = 10$ (higher state space dimensions for the student, namely $d_g = 200$ and $d_g = 500$, yield qualitatively identical results). The student is learned by optimizing an empirical loss comprising training sequences of length $k_g$, where $k_g$ is predetermined. Optimization is based on Adam emanating from standard near-zero initialization. Figure 2(a) reports the extrapolation error (quantified by the $\ell_\infty$ distance between the response of the learned student and that of the teacher, averaged across randomly generated input sequences) for different choices of $k_g$. As can be seen, similarly to the case with linear RNNs (see Sections 5.1 and 5.2), there exists a critical threshold for the training sequence length $k_g$, above which extrapolation error is low and below which extrapolation error is high (note that this critical threshold is around four times the teacher's state space dimension, whereas with linear RNNs the critical threshold was around two times the teacher's state space dimension; theoretically explaining this difference is an interesting direction for future work). Figure 2(b) displays the average output response over different inputs of the teacher alongside those of two students — one trained with

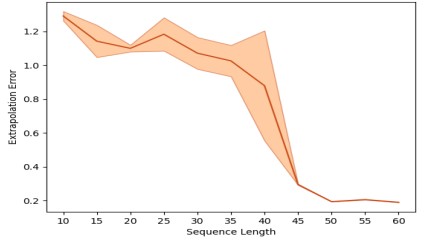

(a) Extrapolation vs. training sequence length

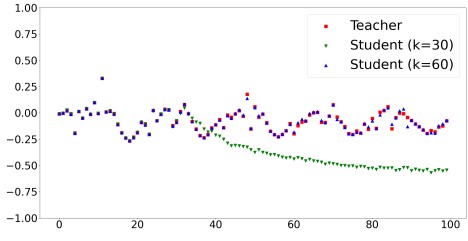

(b) Average output over several inputs

Figure 2: Demonstration of implicit extrapolation with non-linear RNNs, namely GRU networks. Plots show results for a student with state space dimension $d_g = 100$ learning from a teacher with state space dimension $\hat{d}_g = 10$ using training sequences of length $k_g$, where $k_g$ varies. (a) Extrapolation error (average over ten random seeds, with shaded region marking standard deviation) as a function of $k_g$. (b) Average output over several inputs of teacher and student for different choices of $k_g$. Notice that, similarly to the case with linear RNNs, there exists a critical threshold for $k_g$ above which extrapolation error is low and below which extrapolation error is high. See details in Appendix B.

sequences of length $k_g = 30$, and the other with sequences of length $k_g = 60$.[2] As expected, the impulse response of each student tracks that of the teacher for the first $k_g$ time steps (where $k_g$ is student-dependent). However, while the student for which $k_g = 30$ fails to track the teacher beyond $k_g$ time steps, the student for which $k_g = 60$ succeeds, thereby exemplifying implicit extrapolation.

## 6 CONCLUSION

This paper studies the question of extrapolation in RNNs, and more specifically, of whether a student RNN trained on data generated by a teacher RNN can capture the behavior of the teacher over sequences longer than those seen in training. We focus on overparameterized students that can perfectly fit training sequences while producing a wide range of behaviors over longer sequences. Such a student will fail to extrapolate, unless the teacher possesses a certain structure, and the learning algorithm is biased towards solutions adhering to that structure. We show — theoretically for linear RNNs and empirically for both linear and non-linear RNNs — that such implicit extrapolation takes place when the teacher has a low dimensional state space and the learning algorithm is GD.

Existing studies of implicit extrapolation in (linear) RNNs (Emami et al., 2021; Cohen-Karlik et al., 2022) suggest that GD is biased towards solutions with short-term memory. While low dimensional state space and short-term memory may coincide in some cases, in general they do not, and a solution with low dimensional state space may entail long-term memory. Our theory and experiments show that in settings where low dimensional state space and short-term memory contradict each other, the implicit extrapolation chooses the former over the latter.

An important direction for future work is extending our theory to non-linear RNNs. We believe it is possible, in the same way that theories for linear (feed-forward) NNs were extended to account for non-linear NNs (see, e.g., Razin et al. (2021; 2022); Lyu & Li (2019)). An additional direction to explore is the applicability of our results to the recently introduced S4 model (Gu et al., 2022).

## 7 ACKNOWLEDGEMENTS

This work was supported by the European Research Council (ERC) under the European Unions Horizon 2020 research and innovation programme (grant ERC HOLI 819080), the Tel-Aviv University Data-Science and AI Center (TAD), a Google Research Scholar Award, a Google Research Gift, the Yandex Initiative in Machine Learning, the Israel Science Foundation (grant 1780/21), Len Blavatnik and the Blavatnik Family Foundation, and Amnon and Anat Shashua.

---

[2]Note that with GRU networks, in contrast to linear RNNs, the impulse response does not identify the input-output mapping realized by a network. It is presented in Figure 2(b) for demonstrative purposes.

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

## A    NECESSITY OF LOWER BOUND ON TRAINING SEQUENCE LENGTH

Our theoretical guarantees of implicit extrapolation (Theorems 5 and 8) assumed that the training sequence length $k$ is greater than two times the teacher's state space dimension $\hat{d}$. Below we show that this assumption is necessary (up to a small additive constant). More precisely, we prove that if $k \leqslant 2\hat{d} - 1$, implicit extrapolation cannot be guaranteed.

**Lemma A.1.** *For any $\hat{d} \in \mathbb{N}$, there exist two configurations of teacher parameters $\hat{\Theta}_1 = (\hat{A}_1, \hat{B}_1, \hat{C}_1)$ and $\hat{\Theta}_2 = (\hat{A}_2, \hat{B}_2, \hat{C}_2)$, both balanced (Definition 4), stable (meaning the eigenvalues of $\hat{A}_1$ and $\hat{A}_2$ are in $[-1, 1]$) and non-degenerate (meaning the input-output mappings realized by $\hat{\Theta}_1$ and $\hat{\Theta}_2$ are not identically zero), such that:*

$$\hat{B}_1 \hat{A}_1^j \hat{C}_1 = \hat{B}_2 \hat{A}_2^j \hat{C}_2 \ \ for \ all \ \ j = 0, 1, \ldots, 2\hat{d} - 2 \,,$$

*and yet:*

$$\hat{B}_1 \hat{A}_1^j \hat{C}_1 \neq \hat{B}_2 \hat{A}_2^j \hat{C}_2 \ \ for \ \ j = 2\hat{d} - 1 \,.$$

*Proof.* A derivation as in the proof sketch of Lemma 7 shows that any $\hat{d}$-atomic distribution (i.e. any distribution supported on a set of $\hat{d}$ real numbers) can be associated with a balanced configuration of teacher parameters $\hat{\Theta} = (\hat{A}, \hat{B}, \hat{C})$ satisfying $\hat{C}\hat{B} = 1$, such that the values to which the distribution assigns non-zero probability are the eigenvalues of $\hat{A}$, and the $j$th moment of the distribution is equal to $\hat{C}\hat{A}^j\hat{B}$ for every $j \in \mathbb{N}$. In light of this, and of the fact that any configuration of teacher parameters $\hat{\Theta} = (\hat{A}, \hat{B}, \hat{C})$ satisfying $\hat{C}\hat{B} = 1$ is non-degenerate (the first element of its impulse response is non-zero), it suffices to prove that there exist two $\hat{d}$-atomic distributions supported on $[-1, 1]$ which agree on their first $2\hat{d} - 2$ moments yet disagree on their $(2\hat{d} - 1)$'th moment. This follows from Lemmas 4 and 30 in Wu & Yang (2020). $\square$

**Corollary A.2.** *Assume the conditions of Theorem 5, but with $k > 2\hat{d}$ replaced by $k \leqslant 2\hat{d} - 1$. Then, the stated result does not hold, i.e. GF may converge to a point $\Theta^*$ satisfying $\mathcal{L}(\Theta^*) = 0$ which does not extrapolate (Definition 2). Similarly, replacing $k > 2\hat{d}$ by $k \leqslant 2\hat{d} - 1$ in the conditions of Theorem 8 renders the stated result false, meaning there exist $\epsilon > 0$ and $q \in \mathbb{N}$ such that for every $\delta > 0$, there is some $\Theta$ satisfying $\mathcal{L}(\Theta) \leqslant \delta$ which does not $\epsilon$-extrapolate with horizon $q$ (Definition 2).*

*Proof.* In the context of either Theorem 5 or Theorem 8, if $k \leqslant 2\hat{d} - 1$ then by Lemma A.1 there exist two configurations of teacher parameters — both satisfying the conditions of the theorem — which lead to a different impulse response (Definition 1) yet induce the same loss $\mathcal{L}(\cdot)$ (Equation 3.4). If the result stated in the theorem were true, it would mean extrapolation, or $\epsilon$-extrapolation with horizon $q$ for arbitrarily small $\epsilon > 0$ and arbitrarily large $q \in \mathbb{N}$ (see Definition 2), simultaneously with respect to both teachers, and this leads to a contradiction. $\square$

## B    FURTHER EXPERIMENTS

In this section we provide additional experiments that are not included in the main manuscript due to space constraints.

### B.1    BALANCED TEACHER

In Section 5.1, we have experimented with our proposed theoretical setup. In this section we provide additional figures and experiments.

### B.1.1    UNBALANCED STUDENT

In this experiment we use the same balanced teacher with $\hat{d} = 5$ as done in Section 5.1. Instead of the diagonal student with balanced initialization, we use a general (non-diagonal) student with weights sampled from a Gaussian with scale $10^{-5}$ and $d = 40$. Results are depicted in Figure 3(a). A similar phase transition phenomenon to the one in Figure 1 is found also here.

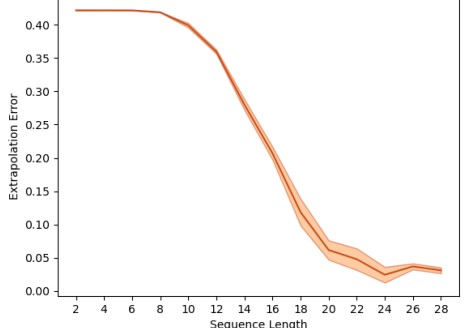 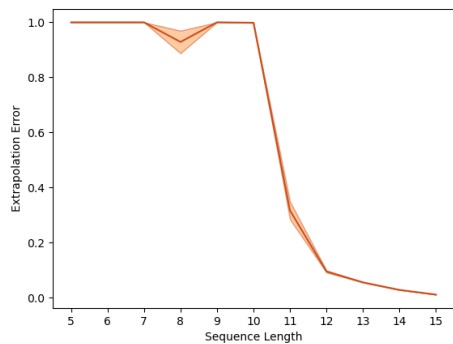

(a) Balanced teacher and general (unbalanced) student (b) Random (unbalanced) teacher and general (unbalanced) student

Figure 3: Extrapolation error as a function of the training sequence length $k$. (a) a balanced teacher with state dimensions $\hat{d} = 5$ and a general (unbalanced and non diagonal) student with $d = 40$. (b) a random unbalanced teacher (see Section B.2) with dimension $\hat{d} = 5$, and a student that has a non-diagonal transition matrix and is trained with standard (small) initialization, with state dimension $d = 50$. In both plots results are averaged over 3 seeds.

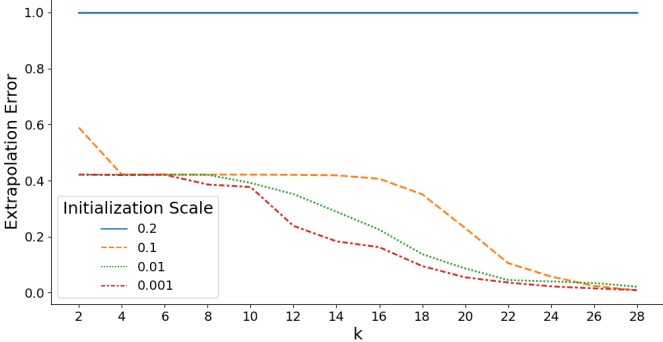

Figure 4: Extrapolation error as a function of training sequence length $k$ for different initialization scales. Extrapolation error increases along with the scale of initialization.

### B.1.2 EFFECT OF THE INITIALIZATION SCALE

Proposition 9 provides theoretical support for the fact that under near-zero initialization, the learned RNN tends to balancedness, which according to theorems 5 and 8 guarantees extrapolation. Below we empirically explore the impact of varying the initialization scale. We use the same setting as in Section B.1.1, and repeat the experiment with different initialization scales for the students' weights.

As can be seen in Figure 4, the extrapolation deteriorates for larger initialization scale, in the sense that it requires longer training sequences for getting good extrapolation error. This suggests that the condition of small initialization required by our theory is not an artifact of our proof technique, but rather a necessary condition for extrapolation to occur.

### B.2 UNBALANCED TEACHER

In Section 5.2, we have tested the extrapolation with respect to a specific unbalanced teacher and have observed a similar phase transition as predicted by the theory of Section 4 and empirical evaluation of Section 5.1. Here we show that the phase transition is not limited to the specific teacher discussed by

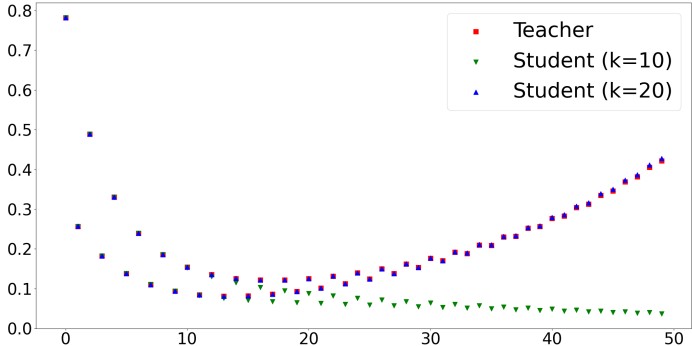

Figure 5: **Balanced teacher and student impulse response**. Students trained with: $k = 10, 20$ with respect to the balanced teacher described in Section C.2.1. As can be seen, both students track the teacher up to the $k$ used in training, for $k = 10$ there is no extrapolation for larger values of $k$, whereas $k = 20$ tracks the teacher well beyond the sequence length used in training.

testing with respect to a randomly generated unbalanced (non-diagonal) teacher (see Section C.2.2). The teacher is set to $\hat{d} = 5$ and student to $d = 50$. Results are presented in Figure 3 (b). Here too we can observe the phase transition phenomena.

### B.3 IMPULSE RESPONSE FIGURES

In Section 5 we have presented the extrapolation performance in different settings. In order to better convey the meaning of extrapolating vs non-extrapolating solutions we present here figures of the impulse response of different models.

We start with the impulse response corresponding to the experiment described in Section 5.1. Figure 5 depicts the balanced teacher with $\hat{d} = 5$ and two selected students (with $d = 40$), one trained with $k = 10$ and the other with $k = 20$.

We can see that the student trained with $k = 10$ tracks the teacher several steps beyond the $10th$ time step and then decays to zero. For $k = 20$ we can see near perfect extrapolation for the horizon evaluated.

Next we turn to Section 5.2 and depict the average impulse responses (Figure 6) of the "delay teacher" and the students trained with respect to the mentioned teacher.

Since the teacher here has $\hat{d} = 10$, a model trained with $k = 8$ is trained with respect to the zero impulse response (see Section C.2.2 for details on delay teacher), and as expected results with the 'zero' solution. we can see that for $k = 18$ the student diverges from the teacher shortly after the $18th$ time step. For $k = 20$ we can see near perfect extrapolation up to the horizon considered.

## C IMPLEMENTATION DETAILS

All the experiments are implemented using PyTorch.

### C.1 OPTIMIZATION

In Section 5.1 we optimize the population loss, which entails minimizing Equation 3.4 with respect to the parameters of the learned model. We use 15K optimization steps with Adam optimizer and a learning rate of $10^{-3}$. In this experiment, the results were not sensitive to the initialization scale of the (balanced) student. In Section 5.2 and Section 5.3 in the experiments that involve minimizing the empirical loss, we use 50K optimization steps with early stopping (most experiments required less than 10K steps). The batch size is set to 100, data is sampled from a Gaussian with zero mean

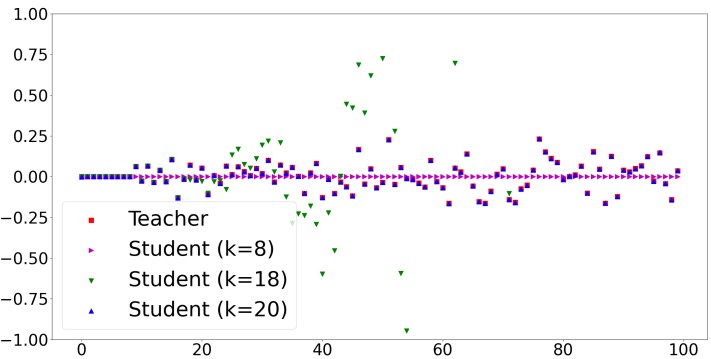

Figure 6: **Unbalanced teacher (delay) and student impulse response**. Students trained with: $k = 8, 18, 20$ with respect to the unbalanced delay teacher described in Section C.2.2. We can see that for $k = 18$ the student diverges for longer sequences while $k = 20$ which is trained for merely two additional time steps extrapolates and tracks the teacher almost perfectly.

and scale of 1. Experiments were not sensitive to most hyper-parameters other than learning rate and initialization scale. The examination of the effect of initialization scale presented in Section B.1.2 is done with learning rate scheduler `torch.optim.lr_scheduler.MultiStepLR` using milestones at $[5000, 10000, 15000, 30000]$ and a decaying factor of $\gamma = 0.1$.

## C.2 TEACHER GENERATION

One of the main challenges in empirically evaluating extrapolation is that randomly sampling weights from a Gaussian distribution may result with an RNN of lower effective rank (i.e. the resulting RNN may be accurately approximated with another RNN with a smaller hidden dimension). We will now describe the teacher generation scheme for the different experiments.

### C.2.1 BALANCED TEACHER GENERATION

A balanced teacher consists of $d$ entries corresponding to the diagonal teacher and $d$ entries representing $\hat{B} = \hat{C}^\top$. In order to avoid cases of rapid decay in the impulse response on the one hand, and exponential growth on the other, we set the eigenvalues to distribute uniformly between $0.6$ and $1.05$. The values of $\hat{B}$ and $\hat{C}$ are randomly sampled from a Gaussian around $0.5$ and scale $1$ and then normalized such that $\hat{C}\hat{B} = 1$.

### C.2.2 UNBALANCED TEACHER GENERATION

In this experiment, the teacher has a general (non-symmetrid) matrix $\hat{A}$ and $\hat{B} \neq \hat{C}^\top$. We set the weights as described next.

**Delay Teacher** A 'delay' teacher has an impulse response of $1$ at time step $i = \hat{d} - 1$, that is, the teacher has an impulse response of $(0, \ldots, 0, 1, 0, \ldots)$. In order to generate the mentioned impulse response we set the weights as follows,

$$A = \begin{pmatrix} 0 & 1 & & 0 \\ & & \ddots & \\ 0 & 0 & & 1 \\ 0 & 0 & & 0 \end{pmatrix}, \ B = \begin{pmatrix} 0 \\ \vdots \\ 0 \\ 1 \end{pmatrix}, \text{ and } C^\top = \begin{pmatrix} 1 \\ 0 \\ \vdots \\ 0 \end{pmatrix}. \tag{C.1}$$

Note that $B, C$ above are set to extract the last entry of the first row of $A^i$ and $A$ is a Nilpotent shift matrix. It is straightforward to verify that $CA^iB = 1$ for $i = \hat{d} - 1$ and $0$ otherwise.

**Random Unbalanced Teacher** The second unbalanced teacher is randomly generated. In order to avoid the caveats mentioned in Section B.1, we randomly sample the diagonal (from a Gaussian with zero mean and scale $0.1$) and super diagonal (from a Gaussian with mean $0.7$ and scale $0.1$) of $A$. We set $B, C$ as in equation C.1. The structure of $\boldsymbol{A}$ ensures similar properties to that of the delayed teacher, specifically, that the first entries of the impulse response is zero and the teacher is 'revealed' only after $\hat{d}$ time steps.

### C.2.3 Non-Linear Teacher Generation

As opposed to the linear teacher discussed in previous sections, when the teacher is a Gated Recurrent Units (GRU), it is unclear how to generate a non-trivial teacher. When randomly generating a teacher GRU the result is either a trivial model that quickly decays to zero or a teacher with an exploding impulse response (depending on the scale of the initialization). In order to produce a teacher with interesting extrapolation behaviour, we initialize a model with an initialization scale of $10^{-6}$ and train for $1000$ step the model to mimic an arbitrarily chosen impulse response. The result of the mentioned procedure is a teacher GRU with non-trivial behaviour. Figure 2(b) shows that we get with this non-trivial teacher the phase transition phenomena as described in Section 5.3.

### C.3 Extrapolation Error

The concept of extrapolation is very intuitive, and yet it does not admit any standard error measure. A proper extrapolation error measure should: (a) capture fine differences between two models with good extrapolation behaviour; and on the other hand, (b) be insensitive to the scale in which two non-extrapolating model explode. A natural approach which we take here is to report the $\ell_\infty$ norm difference on the tail of the impulse response. A model is considered non-extrapolating if the extrapolation error is worse than the extrapolation error of a trivial solution which has an impulse response of zeros.

## D Accumulating Loss

In the main paper the analysis is performed for the loss function defined in Section 3, which corresponds to a regression problem over sequences. Another important and common loss function is an accumulating loss defined over the full output sequence. Specifically, the empirical loss of Equation 3.3 is replaced with,

$$\mathcal{L}_S(\boldsymbol{A}, \boldsymbol{B}, \boldsymbol{C}) = \frac{1}{N} \sum_{i=1}^{N} \sum_{j=0}^{k-1} \ell\left( RNN\left( x_j^{(i)} \right), y_j^{(i)} \right), \tag{D.1}$$

In this section we discuss the adaptations required to accommodate our theory with the loss defined in Equation D.1.

### D.1 Population Loss

A similar derivation of the population loss described in Appendix E.1.1 can be applied to Equation D.1. The difference is that an additional summation is introduced and is preserved throughout the analysis to result with,

$$\mathbb{E}_{\boldsymbol{x} \sim \mathcal{D}} \left[ \sum_{j=0}^{k-1} \ell\left( RNN\left( x_j \right), y_j \right) \right] = \sum_{j=0}^{k-1} \sum_{i=0}^{j} \left( \boldsymbol{C}\boldsymbol{A}^i\boldsymbol{B} - w_i \right)^2. \tag{D.2}$$

The loss above can be viewed as a different weighting of the original population loss, i.e. Equation D.2 can be written as

$$\sum_{i=0}^{k-1} (k-i)(\boldsymbol{C}\boldsymbol{A}^i\boldsymbol{B} - w_i)^2. \tag{D.3}$$

It is clear that the minimizers of Equation D.2 are the same minimizers of Equation 3.4 (i.e. $\boldsymbol{C}\boldsymbol{A}^i\boldsymbol{B} = w_i$ for $i = 0, \ldots, k-1$). Thus Lemma 7 holds with no additional modifications.

## D.2 APPROXIMATE EXTRAPOLATION

For Theorem 8 the analysis in the proof makes use of the fact that the difference in the moments defined by the student and teacher is bounded by $\mathcal{O}(\sqrt{\delta})$. The same is true for the case of the weighted loss, specifically, if the loss $\leqslant \delta$, then for all $i = 0, \ldots, k-1$, $\sqrt{(k-i)}(\boldsymbol{C}\boldsymbol{A}^i\boldsymbol{B} - w_i) \leqslant \sqrt{\delta}$. Since $k - i \geqslant 1$ for $i = 0, \ldots, k-1$ we have $(\boldsymbol{C}\boldsymbol{A}^i\boldsymbol{B} - w_i) \leqslant \sqrt{(k-i)}(\boldsymbol{C}\boldsymbol{A}^i\boldsymbol{B} - w_i) \leqslant \sqrt{\delta}$ and the remainder of the proof is the same.

## D.3 IMPLICIT BIAS FOR BALANCEDNESS

The proof of the implicit bias for balancedness involve the gradients of the population loss defined in Equation 3.4. For the weighted population loss the gradients differ, but the symmetries are all preserved (the gradient computation boils down to adding an external summation to the terms computed in Section E.1.4. The same steps described in Section E.4 apply for the weighted loss.

## E DEFERRED PROOFS

Here we provide complete proofs for the results in the paper.

## E.1 AUXILARY PROOFS

In this section we provide missing proofs from the main paper and additional lemmas to be used in the main proofs.

### E.1.1 POPULATION LOSS

**Lemma E.1 (Proof of Equation 3.4).** *Assume* $\boldsymbol{x} \sim \mathcal{D}$ *such that* $\mathbb{E}_{\boldsymbol{x} \sim \mathcal{D}}[\boldsymbol{x}] = 0, \mathbb{E}_{\boldsymbol{x} \sim \mathcal{D}}[\boldsymbol{x}\boldsymbol{x}^\top] = \boldsymbol{I}_k \in \mathbb{R}^{k,k}$, *where* $\boldsymbol{I}_k$ *is the identity matrix.* $y$ *is given by* $y = \widehat{RNN}(\boldsymbol{x})$ *where* $\widehat{RNN}(\cdot)$ *denotes the output of a teacher RNN,* $\hat{\Theta} = (\hat{\boldsymbol{A}}, \hat{\boldsymbol{B}}, \hat{\boldsymbol{C}})$. *Denote* $w_i = \hat{\boldsymbol{C}}\hat{\boldsymbol{A}}^i\hat{\boldsymbol{B}}$, *the loss for the student RNN satisfies:*

$$\mathbb{E}_{\boldsymbol{x} \sim \mathcal{D}}\left[\ell\left(RNN\left(\boldsymbol{x}\right), y\right)\right] = \sum_{i=0}^{k-1}\left(\boldsymbol{C}\boldsymbol{A}^i\boldsymbol{B} - w_i\right)^2. \tag{E.1}$$

*Proof of Lemma E.1.* The population loss for training with sequences of length $k$ is

$$\mathbb{E}_{\boldsymbol{x} \sim \mathcal{D}}\left[\ell\left(RNN\left(\boldsymbol{x}\right), y\right)\right] = \mathbb{E}_{\boldsymbol{x} \sim \mathcal{D}}\left[\left(\sum_{i=0}^{k-1}\boldsymbol{C}\boldsymbol{A}^{k-1-i}\boldsymbol{B}x_i - \sum_{j=0}^{k-1}w_{k-1-j}x_j\right)^2\right]. \tag{E.2}$$

Reversing the order of summation, expanding the terms,

$$\mathbb{E}_{\boldsymbol{x} \sim \mathcal{D}}\left[\ell\left(RNN\left(\boldsymbol{x}\right), y\right)\right] = \mathbb{E}_{\boldsymbol{x} \sim \mathcal{D}}\left[\left(\sum_{i=0}^{k-1} \boldsymbol{C} \boldsymbol{A}^i \boldsymbol{B} x_{k-1-i} - \sum_{j=0}^{k-1} w_j x_{k-1-j}\right)^2\right] \tag{E.3}$$

$$= \sum_{i,j=0}^{k-1}\left[\boldsymbol{C} \boldsymbol{A}^i \boldsymbol{B} \boldsymbol{C} \boldsymbol{A}^j \boldsymbol{B} - 2 \boldsymbol{C} \boldsymbol{A}^i \boldsymbol{B} w_j + w_i w_j\right] \mathbb{E}_{\boldsymbol{x} \sim \mathcal{D}}\left[x_{k-1-i} x_{k-1-j}\right] \tag{E.4}$$

$$= \sum_{i,j=0}^{k-1}\left[\boldsymbol{C} \boldsymbol{A}^i \boldsymbol{B} \boldsymbol{C} \boldsymbol{A}^j \boldsymbol{B} - 2 \boldsymbol{C} \boldsymbol{A}^i \boldsymbol{B} w_j + w_i w_j\right] \boldsymbol{1}_{\mathrm{k-1-i=k-1-j}} \tag{E.5}$$

$$= \sum_{i,j=0}^{k-1}\left[\boldsymbol{C} \boldsymbol{A}^i \boldsymbol{B} \boldsymbol{C} \boldsymbol{A}^j \boldsymbol{B} - 2 \boldsymbol{C} \boldsymbol{A}^i \boldsymbol{B} w_j + w_i w_j\right] \boldsymbol{1}_{\mathrm{i=j}} \tag{E.6}$$

$$= \sum_{i=0}^{k-1}\left[(\boldsymbol{C} \boldsymbol{A}^i \boldsymbol{B})^2 - 2 \boldsymbol{C} \boldsymbol{A}^i \boldsymbol{B} w_i + w_i^2\right] = \sum_{i=0}^{k-1}\left(\boldsymbol{C} \boldsymbol{A}^i \boldsymbol{B} - w_i\right)^2. \tag{E.7}$$

where the transition from the second to third rows is by our assumption that $\mathbb{E}_{\boldsymbol{x} \sim \mathcal{D}}[x_i x_j] = \boldsymbol{1}_{\mathrm{i=j}}$. Therefore we have,

$$\mathbb{E}_{\boldsymbol{x} \sim \mathcal{D}}\left[\ell\left(RNN\left(\boldsymbol{x}\right), y\right)\right] = \sum_{i=0}^{k-1}\left(\boldsymbol{C} \boldsymbol{A}^i \boldsymbol{B} - w_i\right)^2. \tag{E.8}$$

concluding the proof. $\qquad\square$

### E.1.2   Perfect Generalization and Failed Extrapolation

**Proposition E.2** (Proposition 3 in main paper)**.** *Assume $d > k$, and let $\epsilon \geqslant 0$ and $q \in \{k + 1, k + 2, \ldots\}$. Then, for any teacher parameters $\hat{\Theta}$, there exist student parameters $\Theta$ with which the population loss in Equation 3.4 equals zero, and yet the student does* not *$\epsilon$-extrapolate with horizon q.*

*Proof.* Consider a student, $\Theta$, such that $\boldsymbol{A}$ is symmetric (and therefore has an orthogonal eigendecomposition). Denote $\boldsymbol{A} = \boldsymbol{U} \boldsymbol{\Lambda} \boldsymbol{U}^\top$. The impulse response at time step $i$ can be expressed as $\boldsymbol{C} \boldsymbol{A}^i \boldsymbol{B} = \boldsymbol{C} \boldsymbol{U} \boldsymbol{\Lambda}^i \boldsymbol{U}^\top \boldsymbol{B}$. The latter can be written compactly in matrix form as $\boldsymbol{V} \boldsymbol{g}$ where $\boldsymbol{V}$ is the Vandermonde matrix with $diag(\boldsymbol{\Lambda})$ as its values,

$$\boldsymbol{V} = \begin{pmatrix} 1 & 1 & \ldots & 1 \\ \lambda_1 & \lambda_2 & \ldots & \lambda_d \\ \lambda_1^2 & \lambda_2^2 & \ldots & \lambda_d^2 \\ \vdots & \vdots & & \vdots \\ \lambda_1^{d-1} & \lambda_2^{d-1} & \ldots & \lambda_d^{d-1} \end{pmatrix},$$

and $\boldsymbol{g}$ is defined as $\boldsymbol{g} \equiv (\boldsymbol{C} \boldsymbol{U})^\top \odot \boldsymbol{U}^\top \boldsymbol{B}$.[3] A known result on square Vandermonde matrices is that they are invertible if and only if $\lambda_i \neq \lambda_j$, $\forall i \neq j$. Given a fixed set of distinct values $(\lambda_1, \ldots, \lambda_d)$ and an arbitrary impulse response $\boldsymbol{r} \in \mathbb{R}^d$, in order for the student to generate the impulse response $\boldsymbol{r}$ (i.e. $\boldsymbol{V} \boldsymbol{g} = \boldsymbol{r}$), one can set the coefficient vector, $\boldsymbol{g} = \boldsymbol{V}^{-1} \boldsymbol{r}$ and end up with a symmetric student with $\boldsymbol{r}$ as its impulse response of length $d$.

Consider a teacher RNN, $\hat{\Theta} = (\boldsymbol{A}, \boldsymbol{B}, \boldsymbol{C})$, we can set and the first $k$ entries of $\boldsymbol{r}$ to $r_i = \hat{\boldsymbol{C}} \hat{\boldsymbol{A}}^{i-1} \hat{\boldsymbol{B}}$, $\forall i = \{1, \ldots, k\}$. We are therefore left with $d - k$ degrees of freedom which yields many different students that correspond to the first $k$ entries of the teacher while fitting arbitrary values beyond the $k$ considered. $\qquad\square$

---

[3] Here $\odot$ denotes the Hadamard (elementwise) product.

### E.1.3 EQUIVALENCE BETWEEN BALANCED RNNS WITH SYMMETRIC AND DIAGONAL TRANSITION MATRICES

**Lemma E.3.** *A balanced RNN, $\Theta = (\boldsymbol{A}, \boldsymbol{B}, \boldsymbol{C})$, with a symmetric transition matrix (i.e. $\boldsymbol{B} = \boldsymbol{C}^\top$ and $\boldsymbol{A} = \boldsymbol{A}^\top$) has an equivalent (i.e. generating the same impulse response) RNN, $\Theta' = (\boldsymbol{A}', \boldsymbol{B}', \boldsymbol{C}')$, which is balanced and its transition matrix is diagonal.*

Lemma E.3 allows alternating between systems with symmetric and diagonal matrices. This is useful to simplify the analysis in Section 4.

***Proof of Lemma E.3.*** Any symmetric matrix admits an orthogonal eigendecomposition with real (non-imaginary) eigenvalues. Denote $\boldsymbol{A} = \boldsymbol{U}\boldsymbol{\Lambda}\boldsymbol{U}^\top$. We can define

$$\boldsymbol{A}' = \boldsymbol{\Lambda}, \quad \boldsymbol{B}' = \boldsymbol{U}^\top \boldsymbol{B} \quad \text{and} \quad \boldsymbol{C}' = \boldsymbol{C}\boldsymbol{U},$$

The $i^{th}$ index of the impulse response is given by

$$\boldsymbol{C}\boldsymbol{A}^i\boldsymbol{B} = \boldsymbol{C}\boldsymbol{U}\boldsymbol{\Lambda}^i\boldsymbol{U}^\top\boldsymbol{B} = \boldsymbol{C}'\left(\boldsymbol{A}'\right)^i\boldsymbol{B}'$$

concluding that $\Theta$ and $\Theta'$ have the same impulse response of any length. $\qquad\square$

### E.1.4 GRADIENT DERIVATION

For completeness and Section E.4, we compute the gradients for the general setting.

**Lemma E.4.** *Given the population loss*

$$\mathcal{L}(\boldsymbol{A}, \boldsymbol{B}, \boldsymbol{C}) = \sum_{j=0}^{k-1} \left(\boldsymbol{C}\boldsymbol{A}^j\boldsymbol{B} - \hat{\boldsymbol{C}}\hat{\boldsymbol{A}}^j\hat{\boldsymbol{B}}\right)^2. \tag{3.4 revisited}$$

*Denote $\nabla\ell_i = \boldsymbol{C}\boldsymbol{A}^i\boldsymbol{B} - w_i$, the derivatives of the loss with respect to $\boldsymbol{B}$, and $\boldsymbol{C}$ satisfy:*

$$\frac{\partial\mathcal{L}}{\partial\boldsymbol{B}} = \sum_{i=0}^{k-1} \nabla\ell_i(\boldsymbol{A}^i)^\top\boldsymbol{C}^\top, \tag{E.9}$$

$$\frac{\partial\mathcal{L}}{\partial\boldsymbol{C}} = \sum_{i=0}^{k-1} \nabla\ell_i\boldsymbol{B}^\top\left(\boldsymbol{A}^i\right)^\top. \tag{E.10}$$

***Proof of Lemma E.4.*** Here, we will compute the gradient of the population loss.

Note that for $j \geqslant 0$, the derivative of $\boldsymbol{C}\boldsymbol{A}^j\boldsymbol{B}$ with respect to to $\boldsymbol{B}$ is given by

$$\frac{\partial(\boldsymbol{C}\boldsymbol{A}^j\boldsymbol{B})}{\partial\boldsymbol{B}} = (\boldsymbol{A}^j)^\top\boldsymbol{C}^\top. \tag{E.11}$$

Similarly, the derivative of $\boldsymbol{C}\boldsymbol{A}^j\boldsymbol{B}$ with respect to to $\boldsymbol{C}$ is given by

$$\frac{\partial(\boldsymbol{C}\boldsymbol{A}^j\boldsymbol{B})}{\partial\boldsymbol{C}} = \boldsymbol{B}^\top(\boldsymbol{A}^j)^\top. \tag{E.12}$$

Using these derivatives, we can calculate the derivative of the population loss, (assigning $w_i = \hat{\boldsymbol{B}}\hat{\boldsymbol{A}}^i\hat{\boldsymbol{C}}$),

$$\mathcal{L}(\boldsymbol{A}, \boldsymbol{B}, \boldsymbol{C}) = \mathbb{E}_{\boldsymbol{x}\sim\mathcal{D}}\left[\ell\left(RNN\left(\boldsymbol{x}\right), y\right)\right] = \sum_{i=0}^{k-1} \left(\boldsymbol{C}\boldsymbol{A}^i\boldsymbol{B} - w_i\right)^2. \tag{E.13}$$

Denoting $\nabla\ell_i = \boldsymbol{C}\boldsymbol{A}^i\boldsymbol{B} - w_i$, and noting that $w_i$ is constant (depends on $\hat{\Theta}$), we have for $\boldsymbol{X} \in \{\boldsymbol{B}, \boldsymbol{C}\}$:

$$\frac{\partial\mathcal{L}}{\partial\boldsymbol{X}} = \sum_{i=0}^{k-1} \frac{\partial\left(\boldsymbol{C}\boldsymbol{A}^i\boldsymbol{B} - w_i\right)^2}{\partial\boldsymbol{X}} = \sum_{i=0}^{k-1} \nabla\ell_i\frac{\partial\left(\boldsymbol{C}\boldsymbol{A}^i\boldsymbol{B} - w_i\right)}{\partial\boldsymbol{X}} = \sum_{i=0}^{k-1} \nabla\ell_i\frac{\partial\left(\boldsymbol{C}\boldsymbol{A}^i\boldsymbol{B}\right)}{\partial\boldsymbol{X}}. \tag{E.14}$$

Plugging in Equation E.11 and Equation E.12, we have:

$$\frac{\partial \mathcal{L}}{\partial \boldsymbol{B}} = \sum_{i=0}^{k-1} \nabla \ell_i \frac{\partial \left(\boldsymbol{C}\boldsymbol{A}^i\boldsymbol{B}\right)}{\partial \boldsymbol{B}} = \sum_{i=0}^{k-1} \nabla \ell_i (\boldsymbol{A}^i)^\top \boldsymbol{C}^\top, \tag{E.15}$$

$$\frac{\partial \mathcal{L}}{\partial \boldsymbol{C}} = \sum_{i=0}^{k-1} \nabla \ell_i \frac{\partial \left(\boldsymbol{C}\boldsymbol{A}^i\boldsymbol{B}\right)}{\partial \boldsymbol{C}} = \sum_{i=0}^{k-1} \nabla \ell_i \boldsymbol{B}^\top (\boldsymbol{A}^i)^\top, \tag{E.16}$$

$\square$

### E.1.5 LEMMA 6 (CONSERVATION OF BALANCEDNESS)

**Lemma E.5.** *[Lemma 6 in main paper] When optimizing equation 3.4 with GF emenating from a balanced initialization $\Theta(0)$, the parameters $\Theta(\tau)$ are balanced for all $\tau \in \mathbb{R}_+$.*

We prove the above result by first showing it for GD and then translating the result to GF. The GD result is stated below, and generalizes a result that was shown in Cohen-Karlik et al. (2022) for the memoryless case.

**Lemma E.6.** *When optimizing equation 3.4 with GD with balanced initial conditions, then $\forall t \in \mathbb{N}$, $\Theta$ has a balanced weight configuration, i.e. $\boldsymbol{B}_t = \boldsymbol{C}_t^\top$.*

***Proof of Lemma E.6.*** We prove by induction. By our assumption, the condition holds for $t = 0$. Assume $\boldsymbol{B}_t = \boldsymbol{C}_t^\top$, our goal is to show the conditions hold for $(\boldsymbol{B}_{t+1}, \boldsymbol{C}_{t+1})$. In order to show that $\boldsymbol{B}_{t+1} = \boldsymbol{C}_{t+1}^\top$, we only need to show that $\frac{\partial \mathcal{L}}{\partial \boldsymbol{B}_t} = \left(\frac{\partial \mathcal{L}}{\partial \boldsymbol{C}_t}\right)^\top$. Writing the gradients (Lemma E.4), we have

$$\left(\frac{\partial \mathcal{L}}{\partial \boldsymbol{C}_t}\right)^\top = \sum_{i=0}^{k-1} \nabla \ell_i \boldsymbol{A}_t^i \boldsymbol{B}_t = \sum_{i=0}^{k-1} \nabla \ell_i (\boldsymbol{A}_t^\top)^i \boldsymbol{C}_t^\top = \frac{\partial \mathcal{L}}{\partial \boldsymbol{B}_t}, \tag{E.17}$$

where the inequality follows from the induction assumption and the symmetric structure of $\boldsymbol{A}_t$. To conclude, the gradients at time $t$ are the same and $\boldsymbol{B}_t = \boldsymbol{C}_t^\top$ by the induction assumption, arriving at

$$\boldsymbol{B}_{t+1} = \boldsymbol{B}_t - \eta \frac{\partial \mathcal{L}}{\partial \boldsymbol{B}_t} = \boldsymbol{C}_t^\top - \eta \left(\frac{\partial \mathcal{L}}{\partial \boldsymbol{C}_t}\right)^\top = \boldsymbol{C}_{t+1}^\top \tag{E.18}$$

$\square$

The proof of Lemma 6 follows from Lemma E.6 and the fact that for sufficiently small step size GD approximates GF with arbitrary precision (see Theorem 3 in Elkabetz & Cohen, 2021).

### E.1.6 CONSERVATION OF DIFFERENCE OF NORMS

Appendix E.1.5 shows that if weights are initialized to be balanced, this property is conserved throughout optimization. Here we show under standard initialization schemes, the difference between the norms of $\boldsymbol{B}$ and $\boldsymbol{C}$ is also conserved.

**Lemma E.7.** *When optimizing equation 3.4 with GF the difference between the norms of $\boldsymbol{B}, \boldsymbol{C}$ is conserved throughout GF, i.e.,*

$$\frac{d}{dt}\left(\|\boldsymbol{B}\|_F^2 - \|\boldsymbol{C}\|_F^2\right) = 0. \tag{E.19}$$

***Proof of Lemma E.7.*** We wish to prove that the difference between the norms is conserved over time. Consider the following expression:[4]

$$\alpha \equiv \|\boldsymbol{B}\|_F^2 - \|\boldsymbol{C}\|_F^2 = Tr(\boldsymbol{B}^\top \boldsymbol{B}) - Tr(\boldsymbol{C}\boldsymbol{C}^\top) = \boldsymbol{B}^\top \boldsymbol{B} - \boldsymbol{C}\boldsymbol{C}^\top. \tag{E.20}$$

---

[4]The last equality follows since in the SISO setup, $\boldsymbol{B}^\top \boldsymbol{B}$ and $\boldsymbol{C}\boldsymbol{C}^\top$ are scalars and therefore the trace operator can be omitted.

With this notation, we just need to prove that $\dot{\alpha} = 0$. The derivative of $\boldsymbol{B}, \boldsymbol{C}$ with respect to time is given by,

$$\dot{\boldsymbol{B}} = -\sum_{i=0}^{k-1} \nabla \ell_i (\boldsymbol{A}^\top)^i \boldsymbol{C}^\top, \tag{E.21}$$

$$\dot{\boldsymbol{C}} = -\sum_{i=0}^{k-1} \nabla \ell_i \boldsymbol{B}^\top (\boldsymbol{A}^\top)^i. \tag{E.22}$$

Using the interchangeability of derivative and transpose, we have:

$$\dot{\alpha} = \dot{\boldsymbol{B}}^\top \boldsymbol{B} + \boldsymbol{B}^\top \dot{\boldsymbol{B}} - \dot{\boldsymbol{C}} \boldsymbol{C}^\top - \boldsymbol{C} \dot{\boldsymbol{C}}^\top = 2\boldsymbol{B}^\top \dot{\boldsymbol{B}} - 2\dot{\boldsymbol{C}} \boldsymbol{C}^\top. \tag{E.23}$$

Plugging equation E.21 and equation E.22, we get

$$\dot{\alpha} = 2\boldsymbol{B}^\top \left( -\sum_{i=0}^{k-1} \nabla \ell_i (\boldsymbol{A}^\top)^i \boldsymbol{C}^\top \right) - 2 \left( -\sum_{i=0}^{k-1} \nabla \ell_i \boldsymbol{B}^\top (\boldsymbol{A}^\top)^i \right) \boldsymbol{C}^\top \tag{E.24}$$

$$= -2 \left[ \boldsymbol{B}^\top \left( \sum_{i=0}^{k-1} \nabla \ell_i (\boldsymbol{A}^\top)^i \right) \boldsymbol{C}^\top - \boldsymbol{B}^\top \left( \sum_{i=0}^{k-1} \nabla \ell_i (\boldsymbol{A}^\top)^i \right) \boldsymbol{C}^\top \right] = 0. \tag{E.25}$$

establishing that $\frac{d}{dt} \left( \|\boldsymbol{B}\|_F^2 - \|\boldsymbol{C}\|_F^2 \right) = 0$.

$\square$

## E.2 LEMMA 7 (EXACT EXTRAPOLATION)

**Lemma E.8.** *[Lemma 7 in main paper] Suppose that $d > k > 2\hat{d}$, the teacher is balanced, and that the student parameters $\Theta$ are balanced and satisfy $\mathcal{L}(\Theta) = 0$. Then $\Theta$ extrapolates.*

***Proof of Lemma E.8.*** By Lemma E.3, a balanced RNN with symmetric transition matrix has an equivalent (generating the same impulse response) balanced RNN with a diagonal transition matrix. We will continue under the assumption of diagonal transition matrices.

Without loss of generality we assume $\hat{\boldsymbol{C}}\hat{\boldsymbol{B}} = 1$. Otherwise, the problem can be rescaled by $\hat{\boldsymbol{C}}\hat{\boldsymbol{B}}$, which is equivalent to rescaling the initial conditions, and providing no additional information.[5]

From the balanced assumption, we have $\hat{\boldsymbol{C}}^\top = \hat{\boldsymbol{B}}$. Denote $\hat{\boldsymbol{p}} = \hat{\boldsymbol{C}}^\top \odot \hat{\boldsymbol{B}} = \hat{\boldsymbol{B}} \odot \hat{\boldsymbol{B}}$, and we get $\hat{p}_i \geqslant 0$ and $\sum_i \hat{p}_i = 1$, and therefore $\hat{\boldsymbol{p}}$ may be interpreted as a distribution over a random variable with $\hat{d}$ possible values. We shall assume that these values are $\hat{A}_{1,1}, \ldots, \hat{A}_{\hat{d},\hat{d}}$, and denote the corresponding random variable by $\hat{Z}$.

Furthermore, we can also interpret elements of the impulse response of $\hat{\Theta}$ as moments of this distribution. Let us write the $n^{th}$ element of the impulse response as:

$$\hat{\boldsymbol{C}}\hat{\boldsymbol{A}}^n \hat{\boldsymbol{B}} = \sum_i \hat{p}_i \hat{a}_i^n = \mathbb{E}_{\hat{\boldsymbol{p}}}[\hat{Z}^n], \tag{E.26}$$

where $\mathbb{E}_{\boldsymbol{p}}[Z]$ is the expected value of a random variable $Z$ under the distribution $\boldsymbol{p}$. In the same way, we can define for the learned model $\Theta$, a distribution $p_i = C_i B_i$, and write the learned impulse response as:

$$\boldsymbol{C}\boldsymbol{A}^n \boldsymbol{B} = \sum_i p_i a_i^n = \mathbb{E}_{\boldsymbol{p}}[Z^n]. \tag{E.27}$$

This view provides us with a moment matching interpretation of the learning problem. Namely, the fact that $\Theta$ matches the first $k$ elements of the teacher impulse response, is the same as saying they agree on the first $k-1$ moments $\mathbb{E}_{\boldsymbol{p}}[Z^j]$ for $j \in \{1, \ldots, k-1\}$.[6] The question of extrapolation is whether equality in the first $k-1$ moments implies an equality in all other moments.

---

[5]The case for which $\hat{\boldsymbol{C}}\hat{\boldsymbol{B}} = 0$ is handled separately.
[6]Equality of the $0^{th}$ moment ensures the student induces a valid probability, i.e. $\sum_i C_i B_i = \sum_i \hat{C}_i \hat{B}_i = 1$.

In (Cohen & Yeredor, 2011, Theorem 1) and in (Wu & Yang, 2020, Lemma 4) it is shown that the first $2\hat{d}$ moments of a discrete random variable taking at most $\hat{d}$ different values uniquely define this random variable. Therefore, any other discrete random variable identifying with the teacher on $2\hat{d}$ moments must be the same random variable and therefore identifies on higher moments as well. Since we assumed $k > 2\hat{d}$, this result immediately implies that equality in the first $k - 1$ moments implies equality in all other moments.

For the case $\hat{C}\hat{B} = 0$, from our assumption that the teacher is balanced, we have that the condition is met only if $\hat{C}_i = \hat{B}_i = 0$ for $i = 1, \ldots, \hat{d}$. Such a teacher has an impulse response of zeros, for $k \geqslant 1$, a student minimizing the loss must also satisfy $CB = 0$ and therefore has the zeros as its impulse response (recall the student is balanced) thus extrapolating with respect to the said teacher.

$\square$

### E.3 THEOREM 8 (APPROXIMATE EXTRAPOLATION)

This section is devoted to the proof of Theorem 8 which ties the approximation error of optimization to that of extrapolation.

**Theorem E.9.** *[Theorem 8 in main paper] Consider the minimization of Equation 3.4 and assume: (i) $d > k > 2\hat{d}$; (ii) the teacher is balanced and stable (i.e. the eigenvalues of $\hat{A}$ are in $[-1, 1]$); (iii) the teacher is non-degenerate, i.e. the input output mapping they realize is not identically zero; (iv) the student parameters are learned by applying GF to the loss $\mathcal{L}(\cdot)$, starting from a balanced initialization; (v) the student parameters $\Theta$ are bounded.*

*Then, for any $\epsilon > 0$ and $q \in \mathbb{N}$, there exists $\delta(\epsilon, q) > 0$ such that whenever $\mathcal{L}(\Theta) \leqslant \delta(\epsilon, q)$, the student $\epsilon$-extrapolates with horizon $q$.*

***Proof of Theorem E.9.*** Let $\delta > 0$ be a constant whose value will be chosen later, and suppose GF reached a point $\Theta$ satisfying $\mathcal{L}(\Theta) \leqslant \delta$. Following the proof of Lemma 7, $\hat{\Theta}$ is identified with a distribution supported on the eigenvalues of $\hat{A}$, whose $j$'th moment is $\hat{m}_j := \hat{C}\hat{A}^j\hat{B}(\hat{C}\hat{B})^{-1}$ for every $j \in \mathbb{N}$. Similarly, $\Theta$ is identified with a distribution supported on the eigenvalues of $A$, whose $j$'th moment is $m_j := CA^jB(CB)^{-1}$ for every $j \in \mathbb{N}$. From our assumption that $\mathcal{L}(\Theta) \leqslant \delta$,

$$\mathcal{L}(\Theta) = \sum_{j=0}^{k-1} \left( CA^jB - \hat{C}\hat{A}^j\hat{B} \right)^2 \leqslant \delta. \tag{E.28}$$

and specifically, each term satisfies $(CA^jB - \hat{C}\hat{A}^j\hat{C})^2 \leqslant \delta$ for $j = 0, \ldots, k-1$. In particular, $(CB - \hat{C}\hat{B})^2 \leqslant \delta$. Denote $\beta = \hat{C}\hat{B} - CB$, then $\beta \in [-\sqrt{\delta}, \sqrt{\delta}]$. Note that $\hat{C}\hat{B}$ is a (positive) constant, multiplying the loss by $(\hat{C}\hat{B})^{-2}$ we have that each term $\leqslant \delta(\hat{C}\hat{B})^{-2}$. We can write for each $j = 0, \ldots, k-1$,

$$\left( \frac{CA^jB}{\hat{C}\hat{B}} - \frac{\hat{C}\hat{A}^j\hat{B}}{\hat{C}\hat{B}} \right)^2 = \left( \frac{CA^jB}{\hat{C}\hat{B}} - \frac{CA^jB}{CB} + \frac{CA^jB}{CB} - \frac{\hat{C}\hat{A}^j\hat{B}}{\hat{C}\hat{B}} \right)^2 \tag{E.29}$$

$$= \left( CA^jB \left( \frac{1}{\hat{C}\hat{B}} - \frac{1}{CB} \right) + \underbrace{\frac{CA^jB}{CB} - \frac{\hat{C}\hat{A}^j\hat{B}}{\hat{C}\hat{B}}}_{m_j - \hat{m}_j} \right)^2 \tag{E.30}$$

We can further expand the term on the left,

$$\frac{1}{\hat{C}\hat{B}} - \frac{1}{CB} = \frac{1}{\hat{C}\hat{B}} - \frac{1}{\hat{C}\hat{B} - \beta} = \frac{\hat{C}\hat{B} - \beta - \hat{C}\hat{B}}{\hat{C}\hat{B}(\hat{C}\hat{B} - \beta)} = \frac{\beta}{\hat{C}\hat{B}(\beta - \hat{C}\hat{B})} \tag{E.31}$$

Plugging back to the above, we have

$$\delta(\hat{C}\hat{B})^{-2} \geqslant \left(\frac{CA^jB}{\hat{C}\hat{B}} - \frac{\hat{C}\hat{A}^j\hat{B}}{\hat{C}\hat{B}}\right)^2 = \left(\frac{\beta CA^jB}{\hat{C}\hat{B}(\beta - \hat{C}\hat{B})} + (m_j - \hat{m}_j)\right)^2 \tag{E.32}$$

$$= \beta^2\kappa^2 + 2\beta\kappa(m_j - \hat{m}_j) + (m_j - \hat{m}_j)^2 \tag{E.33}$$

$$\geqslant 2\beta\kappa(m_j - \hat{m}_j) + (m_j - \hat{m}_j)^2 \tag{E.34}$$

$$\geqslant -2|\delta\kappa(m_j - \hat{m}_j)| + (m_j - \hat{m}_j)^2 \tag{E.35}$$

where $\kappa \equiv \frac{CA^jB}{\hat{C}\hat{B}(\beta - \hat{C}\hat{B})}$. From assumption (ii), the teacher is stable and therefore $\hat{m}_j \leqslant \hat{C}\hat{B}$ for all $j = 0, \ldots, k-1$. Similarly, from assumption (v) the student parameters are bounded and therefore $CA^jB$ is bounded by $\tau^{j+2}$ (where $\tau \equiv \max\{1, \eta\}$ and $\eta$ is a bound on the Frobenous norm of $A, B, C$). $m_j$ is bounded in a similar fashion by $\tau^j$.

Combining the above, for $\delta < \hat{C}\hat{B}$ we have,

$$\frac{\delta}{(\hat{C}\hat{B})^2} \geqslant \frac{-2\delta\tau^{j+2}(\tau^j + 1)}{2(\hat{C}\hat{B})^2} + (m_j - \hat{m}_j)^2 \tag{E.36}$$

Setting $\delta' < \frac{\delta(\hat{C}\hat{B})^2}{1 + \tau^{k+1}(\tau^{k-1}+1)}$, if $\mathcal{L}(\Theta) \leqslant \delta'$ then $|m_j - \hat{m}_j| \leqslant \sqrt{\delta}$ for $j = 1, \ldots, k-1$. Proposition 2 in Wu & Yang (2020) then implies $\mathcal{W}_1(\Theta, \hat{\Theta}) \leqslant \mathcal{O}(\delta^{1/4\hat{d}})$.[7]

Denote $\Omega \equiv \left(\bigcup_{i=1}^d A_{ii}\right) \bigcup \left(\bigcup_{j=1}^{\hat{d}} \hat{A}_{jj}\right)$ (the union of the supports of $\Theta$ and $\hat{\Theta}$), from Section 2.3 in Panaretos & Zemel (2019), for $q > p$ the $q^{th}$ and $p^{th}$ Wasserstein distances satisfy $\mathcal{W}_q^q(\Theta, \hat{\Theta}) \leqslant \mathcal{W}_p^p(\Theta, \hat{\Theta})\gamma^{q-p}$ where $\gamma = \max_{x,y\in\Omega}|x-y|$. In particular, for $p = 1$, $\mathcal{W}_q(\Theta, \hat{\Theta}) \leqslant \left(\mathcal{W}_1(\Theta, \hat{\Theta})\gamma^{q-1}\right)^{1/q}$. Note that $\gamma$ can is bounded by $\gamma \leqslant \tau + 1 \leqslant 2\tau$ (recall the student is bounded and teacher is stable).

Finally, $|m_q - \hat{m}_q| \leqslant \mathcal{W}_q(\Theta, \hat{\Theta})$ (see Section 1.2 in Biswas & Mackey (2021)). Combining the steps above, for all $q \in \mathbb{N}$,

$$|m_q - \hat{m}_q| \leqslant \mathcal{W}_q(\Theta, \hat{\Theta}) \leqslant \left(\mathcal{W}_1(\Theta, \hat{\Theta})(2\tau)^{q-1}\right)^{1/q} \leqslant \left(\rho\delta^{1/4\hat{d}}(2\tau)^{q-1}\right)^{1/q} \tag{E.37}$$

where $\rho$ is a constant satisfying $\mathcal{W}_1(\Theta, \hat{\Theta}) \leqslant \rho\delta^{1/4\hat{d}}$. To achieve $|m_j - \hat{m}_j| < \epsilon$ for any $\epsilon > 0$, we can set $\delta(\epsilon, q) < \left(\frac{\epsilon^q}{\rho\gamma^{q-1}}\right)^{4\hat{d}}$ concluding the proof.

$\square$

### E.4 PROPOSITION 9 (IMPLICIT BIAS FOR BALANCEDNESS)

The proof of Proposition 9 consists of several steps. First, we bound with high probability the norms of $B$ and $C$ at initialization (Lemma E.10). We then derive bounds on the differential equations of $\frac{d}{dt}(B(t) + C^\top(t))$ and $\frac{d}{dt}(B(t) - C^\top(t))$ (Lemma E.14). We show that when the initialization scale tends to zero, the ratio between the differential equations tends to zero. (Lemma E.13).

Before we turn to prove Proposition 9, we first need to bound the initial values for a vector $v \in \mathbb{R}^n$ initialized with $\mathcal{N}(0, \frac{\epsilon^2}{n})$.

**Lemma E.10.** *Assume a vector $v \in \mathbb{R}^n$ with $\mathcal{N}(0, \frac{\epsilon^2}{n})$ per coordinate. Then:*

$$Pr\left(\frac{\epsilon}{2} < \|\mathbf{v}\| < \frac{3\epsilon}{2}\right) \geqslant 1 - 2\exp(-9n/64). \tag{E.38}$$

---

[7]Here we overload notations and denote the distributions of the teacher and student by $\Theta$ and $\hat{\Theta}$ respectively

***Proof of Lemma E.10.*** The proof of E.10 uses known results on the Chi-square distribution Laurent & Massart (2000), applied to our specific setting to achieve the desired bounds. We will begin by changing variables, $\tilde{v}_i = v_i \cdot \frac{\sqrt{n}}{\epsilon}$. The entries $\tilde{v}_i$, are standard Gaussian variables. The squared norm of $\tilde{\boldsymbol{v}}$ distributes according to the $\chi$-squared distribution.

By (Laurent & Massart, 2000, Lemma 1), in our case (assigning $x = 9n/64$) the following inequalities hold:

$$Pr\left(\|\tilde{\boldsymbol{v}}\|^2 \geqslant (1.75 + 9/32)n\right) \leqslant \exp(-9n/64), \tag{E.39}$$

$$Pr\left(\|\tilde{\boldsymbol{v}}\|^2 \leqslant n/4\right) \leqslant \exp(-9n/64). \tag{E.40}$$

In particular,

$$Pr\left(\|\tilde{\boldsymbol{v}}\|^2 \geqslant 2.25n\right) \leqslant \exp(-9n/64) \implies Pr\left(\|\tilde{\boldsymbol{v}}\| \geqslant 1.5\sqrt{n}\right) \leqslant \exp(-9n/64). \tag{E.41}$$

Changing variables back to $\boldsymbol{v}$,

$$Pr\left(\|\boldsymbol{v}\| \geqslant 1.5\epsilon\right) \leqslant \exp(-9n/64). \tag{E.42}$$

Similarly, for the second bound:

$$Pr\left(\|\tilde{\boldsymbol{v}}\| \leqslant \sqrt{n}/2\right) \leqslant \exp(-9n/64) \implies Pr\left(\|\boldsymbol{v}\| \leqslant \epsilon/2\right) \leqslant \exp(-9n/64). \tag{E.43}$$

Taking the complementary probability, we have the desired result of

$$Pr\left(\frac{\epsilon}{2} < \|\boldsymbol{v}\| < \frac{3\epsilon}{2}\right) \geqslant 1 - 2\exp(-9n/64). \tag{E.44}$$

$\square$

Note that for a matrix $\boldsymbol{X} \in \mathbb{R}^{m \times p}$, Lemma E.10 bounds its Frobenius norm, $Pr\left(\frac{\epsilon}{2} < \|\boldsymbol{X}\|_F < \frac{3\epsilon}{2}\right) \geqslant 1 - 2exp\left(-9mp/64\right)$. The result is straight forward by applying the lemma to $\boldsymbol{X}$'s vectorized form.

**Proposition E.11.** *[Proposition 9 in main paper] Suppose that:* (i) $d > 4$; (ii) *the teacher parameters $\hat{\Theta}$ are balanced and are non-degenerate, in the sense that the input-output mapping they realize is not identically zero; and* (iii) *the student parameters are learned by applying GF to the loss $\mathcal{L}(\cdot)$. Let $\tilde{\Theta}$ be a random point in parameter space, with entries drawn independently from the standard normal distribution. For $\epsilon > 0$, consider the case where GF emanates from the initialization $\epsilon\tilde{\Theta}$, and denote the resulting curve by $\Theta_\epsilon(\tau) = (\boldsymbol{A}_\epsilon(\tau), \boldsymbol{B}_\epsilon(\tau), \boldsymbol{C}_\epsilon(\tau))$, with $\tau \geqslant 0$. Then, w.p. at least $0.75$, for every $\epsilon > 0$ there exists $\tau_\epsilon \geqslant 0$ such that:*

$$\lim_{\epsilon \to 0^+} \frac{||\boldsymbol{B}_\epsilon(\tau_\epsilon) - \boldsymbol{C}_\epsilon^\top(\tau_\epsilon)||_F}{||\boldsymbol{B}_\epsilon(\tau_\epsilon) + \boldsymbol{C}_\epsilon^\top(\tau_\epsilon)||_F} = 0. \tag{E.45}$$

The consequence of Proposition 9 is that as $\epsilon$ converges to zero, $\boldsymbol{B}$ and $\boldsymbol{C}$ converge towards each other.

For convenience, we refer to the mentioned initialization scheme (where every coordinate in a vector is initialized as $\mathcal{N}\left(0, \frac{\epsilon^2}{d}\right)$) as $\epsilon$-***normal initialization***. In order to prove the proposition we define a few relevant terms,

$$\boldsymbol{Y} = \boldsymbol{B} - \boldsymbol{C}^\top, \qquad \boldsymbol{W} = \boldsymbol{B} + \boldsymbol{C}^\top, \qquad w_0 = \hat{\boldsymbol{C}}\hat{\boldsymbol{B}}. \tag{E.46}$$

We will in fact prove the stronger, following lemma, for any matrix $\boldsymbol{A}$, not necessarily symmetric.

**Lemma E.12.** *Assume $w_0 > 0$, and $\boldsymbol{A}, \boldsymbol{B}, \boldsymbol{C}$ are $\epsilon$-normally initialized. Then $\exists t$ such that*

$$\lim_{\epsilon \to 0} \frac{\|\boldsymbol{Y}(t)\|^2}{\|\boldsymbol{W}(t)\|^2} = 0, \qquad \lim_{\epsilon \to 0} \frac{\|\boldsymbol{A}(t)\|_F^2}{\|\boldsymbol{W}(t)\|^2} = 0. \tag{E.47}$$

The proof of Lemma E.12 follows three steps: (1) establish a time in the optimization for which the norms of all parameters are bounded (Lemma E.13); (2) derive upper (and lower) bounds for the differential equations describing the evolvement of $\boldsymbol{Y}, \boldsymbol{W}$ and $\boldsymbol{A}$. Our approximations are limited to the initial phase of training. Concretely, we show that for $0 \leqslant t \leqslant \frac{1}{2w_0} \ln\left(\frac{1}{\epsilon^{0.5}}\right)$, all norms are bounded. Thus, it is possible to obtain meaningful bounds on the ODEs of $\boldsymbol{Y}$ and $\boldsymbol{W}$ while $\boldsymbol{A}$ remains in the magnitude of initialization (Lemma E.14); (3) using the relevant bounds, we show that as the initialization scale tends to zero, so do the limits in Equation E.47.

As it turns out, there is a critical time $\bar{t} = \mathcal{O}\left(\ln\left(\frac{1}{\epsilon^{0.5}}\right)\right)$, up until which the considered bounds are valid (see details in the proof of Lemma E.13).

**Lemma E.13.** *Assume $d > 20$, student parameters are $\epsilon$-normally initialized, assume also a balanced teacher. Then w.p. at least 0.75, for all $0 \leqslant t \leqslant \bar{t}$, there exist $M_1, M_2$ such that:*

$$\|\boldsymbol{C}(t)\|, \|\boldsymbol{B}(t)\| < M_1 \epsilon^{0.75} \tag{E.48}$$

*and*

$$\|\boldsymbol{A}(t)\|_F < M_2 \epsilon \tag{E.49}$$

To prove this, we note that at initialization, $\boldsymbol{A}, \boldsymbol{B}$ and $\boldsymbol{C}$ satisfy these bounds. From continuity, there exists a maximal time for which they are satisfied. We bound the rate of their growth, and thus show that for all $t$ as described, we are within this region.

**Lemma E.14.** *Assume $w_0 > 0$ (see Lemma E.1 for definition of $w_i$) and assume $\boldsymbol{A}, \boldsymbol{B}, \boldsymbol{C}$ are $\epsilon$-normally initialized, we have the following bounds hold for all $0 \leqslant t \leqslant \bar{t}$ w.p. at least 0.75,*

$$\boldsymbol{Y}(t)^\top \boldsymbol{Y}(t) \leqslant c_1 \epsilon^2 e^{-2w_0 t} + c_2 \epsilon^{2.5}, \tag{E.50}$$

*and*

$$\boldsymbol{W}(t)^\top \boldsymbol{W}(t) \geqslant c_3 \epsilon^2 e^{2w_0 t} - c_4 \epsilon^{2.5}. \tag{E.51}$$

Lemma E.14 shows that the growth rate of $\boldsymbol{W}(t)^\top \boldsymbol{W}(t)$ and the decay rate of $\boldsymbol{Y}(t)^\top \boldsymbol{Y}(t)$ both depend on the sign of $w_0$. In our analysis we assume the teacher is balanced and therefore $w_0 > 0$, the same analysis applies for $w_0 < 0$ with opposite roles for $\boldsymbol{Y}$ and $\boldsymbol{W}$. The proof of Lemma E.14 follows from writing the leading terms of the ODE and bounding the remaining terms by their upper bounds in the time considered. Using these lemmas, we proceed to prove Lemma E.12.

***Proof of Lemma E.12.*** Consider the dynamics at time $\bar{t} = C \ln\left(\frac{1}{\epsilon^{0.5}}\right)$. By Lemma E.14, w.p. at least 0.75, we have,

$$\boldsymbol{Y}(t)^\top \boldsymbol{Y}(t) \leqslant c_1 \epsilon^2 e^{-2w_0 C \ln \frac{1}{\epsilon^{0.5}}} + c_2 \epsilon^{2.5} = (c_1 e^{-2w_0 C} + c_2)\epsilon^{2.5} \tag{E.52}$$

and

$$\boldsymbol{W}(t)^\top \boldsymbol{W}(t) \geqslant c_3 \epsilon^2 e^{2w_0 C \ln \frac{1}{\epsilon^{0.5}}} - c_4 \epsilon^{2.5} = c_3 e^{2w_0 C} \epsilon^{1.5} - c_4 \epsilon^{2.5} \tag{E.53}$$

We can calculate the limit (where $\tilde{c}_1$ and $\tilde{c}_3$ account for the relevant constant factors),

$$\lim_{\epsilon \to 0} \frac{\|\boldsymbol{Y}(t)\|^2}{\|\boldsymbol{W}(t)\|^2} \leqslant \lim_{\epsilon \to 0} \frac{(\tilde{c}_1 + c_2)\epsilon^{2.5}}{\tilde{c}_3 \epsilon^{1.5} - c_4 \epsilon^{2.5}} = 0 \tag{E.54}$$

From Lemma E.13, $\|\boldsymbol{A}(\bar{t})\|_F \leqslant M_2 \epsilon$, so we can calculate the limit,

$$\lim_{\epsilon \to 0} \frac{\|\boldsymbol{A}(\bar{t})\|_F^2}{\|\boldsymbol{W}(\bar{t})\|^2} \leqslant \lim_{\epsilon \to 0} \frac{M_2 \epsilon^2}{\tilde{c}_3 \epsilon^{1.5} - c_4 \epsilon^{2.5}} = 0 \tag{E.55}$$

which concludes the proof. $\qquad\square$

***Proof of Lemma E.13.*** Applying Lemma E.10 with $d > 20$ results with the bounds holding at initialization with probabilities $\geqslant 1 - 2exp(-9 \cdot 20^2/64)$ for $\boldsymbol{A}$, and $\geqslant 1 - 2exp(-9 \cdot 20/64)$ for $\boldsymbol{B}$ and $\boldsymbol{C}$. The probability for $\boldsymbol{A}, \boldsymbol{B}, \boldsymbol{C}$ satisfying the inequalities simultaneously $\geqslant (1 - 2exp(-9 \cdot 20/64))^3 \approx 0.83 > 0.75$.

Suppose that the norm bounds of Equation E.38 are satisfied at $t = 0$. In particular, $\exists M_1, M_2$ such that

$$\|\boldsymbol{B}(0)\|, \|\boldsymbol{C}(0)\| < 2\epsilon < M_1 \epsilon^{0.75}, \tag{E.56}$$

and

$$\|\boldsymbol{A}(0)\|_F < 2\epsilon < M_2\epsilon. \tag{E.57}$$

Where $2 < M_2 < \frac{1}{\epsilon}$, and $M_1 > 4\epsilon^{0.25}$.

Denote by $t_A$ the minimal time for which $\|\boldsymbol{A}(t_A)\| = M_2\epsilon$. Similarly, $t_B, t_C$ are the times for which $\|\boldsymbol{B}(t_B)\| = \|\boldsymbol{C}(t_C)\| = M_1\epsilon^{0.75}$. Denote also $\bar{t} = \min\{t_A, t_B, t_C\}$. Proving the lemma amounts to showing there exists $C \in \mathbb{R}$ such that $\bar{t} = C \ln\left(\frac{1}{\epsilon^{0.5}}\right)$. Next, we turn to develop the differential inequalities of the norms, which will later be used to lower bound the time until violation of the mentioned bounds.

Recall the derivative of $\boldsymbol{B}$ with respect to time (see Section E.1.4),

$$\dot{\boldsymbol{B}} = -\sum_{i=0}^{k-1} \nabla\ell_i(\boldsymbol{A}^\top)^i \boldsymbol{C}^\top. \tag{E.58}$$

Using Cauchy-Schwartz inequality, we have that for all $t \in [0, \bar{t}]$, the norm of $\boldsymbol{B}$ is upper bounded by

$$\left\|\dot{\boldsymbol{B}}\right\| = \left\|-\sum_{i=0}^{k-1} \nabla\ell_i(\boldsymbol{A}^\top)^i \boldsymbol{C}^\top\right\| \leqslant \sum_{i=0}^{k-1} |\nabla\ell_i| \left\|\boldsymbol{A}^\top\right\|_F^i \left\|\boldsymbol{C}^\top\right\|. \tag{E.59}$$

We now bound the norms of $\nabla\ell_i$, $\boldsymbol{A}$ and $\boldsymbol{C}$ in order to transfer the inequality to a differential one.

Denote $M = \max_i(|w_i|) + M_1^2\epsilon^{1.5}$, then we have

$$M = \max_i(|w_i|) + M_1^2\epsilon^{1.5} \geqslant \max_i(|w_i|) + \|\boldsymbol{C}\|\|\boldsymbol{B}\| \geqslant \max_i(|w_i|) + |\boldsymbol{C}\boldsymbol{B}|, \tag{E.60}$$

$$\geqslant \max_i(|w_i|) + \max_i(|\boldsymbol{C}\boldsymbol{A}^i\boldsymbol{B}|) \geqslant \max_i(|w_i| + |\boldsymbol{C}\boldsymbol{A}^i\boldsymbol{B}|) \geqslant \max_i(|\nabla\ell_i|). \tag{E.61}$$

For the norm of $\boldsymbol{C}$, recall the conservation law from Lemma E.7 for the norms of $\boldsymbol{B}$ and $\boldsymbol{C}$:

$$\forall t, \ \frac{d}{dt}(\|\boldsymbol{B}(t)\| - \|\boldsymbol{C}(t)\|) = 0. \tag{E.62}$$

From the assumption that the initial conditions are met,

$$\|\boldsymbol{B}(0)\|, \|\boldsymbol{C}(0)\| < 2\epsilon \Rightarrow (|\|\boldsymbol{B}(0)\| - \|\boldsymbol{C}(0)\||) < 4\epsilon. \tag{E.63}$$

Therefore, we get

$$\forall t, \ \|\boldsymbol{C}(t)\| < \|\boldsymbol{B}(t)\| + 4\epsilon. \tag{E.64}$$

Note also that by assuming $M_2 < \frac{1}{\epsilon}$, we have $M_2\epsilon < 1$ and

$$\sum_{i=0}^{k-1} (M_2\epsilon)^i < k. \tag{E.65}$$

Plugging the above steps into equation E.59, we have:

$$\|\dot{\boldsymbol{B}}\| \leqslant \sum_{i=0}^{k-1} |\nabla\ell_i|\|(\boldsymbol{A}^\top)\|_F^i\|\boldsymbol{C}^\top\| < M(\|\boldsymbol{B}\| + 4\epsilon) \sum_{i=0}^{k-1} \|(\boldsymbol{A}^\top)\|_F^i \tag{E.66}$$

$$< Mk(\|\boldsymbol{B}\| + 4\epsilon). \tag{E.67}$$

Denoting $\gamma = \|\boldsymbol{B}\|^2 = \boldsymbol{B}^\top\boldsymbol{B}$, then

$$\dot{\gamma} = \dot{\boldsymbol{B}}^\top\boldsymbol{B} + \boldsymbol{B}^\top\dot{\boldsymbol{B}} = 2\boldsymbol{B}^\top\dot{\boldsymbol{B}} \tag{E.68}$$

Taking absolute value and then plugging equation E.66 results with

$$|\dot{\gamma}| = |2\boldsymbol{B}^\top\dot{\boldsymbol{B}}| \leqslant 2\|\boldsymbol{B}^\top\|\|\dot{\boldsymbol{B}}\| < 2\|\boldsymbol{B}\|Mk(\|\boldsymbol{B}\| + 4\epsilon). \tag{E.69}$$

Using the definition of $\gamma$, we get that

$$|\dot{\gamma}| < 2kM\left(|\gamma| + 4\epsilon\sqrt{|\gamma|}\right). \tag{E.70}$$

Next we show that $\bar{t} = O\left(\ln\left(\frac{1}{\epsilon^{0.5}}\right)\right)$. Suppose this is not the case, then there exists $\tilde{t} < \ln\left(\frac{1}{\epsilon^{0.5}}\right)$ such that one of the bounds are violated: (i) $\|\boldsymbol{B}(\tilde{t})\| \geqslant M_1\epsilon^{0.75} > 4\epsilon$; (ii) $\|\boldsymbol{C}(\tilde{t})\| \geqslant M_1\epsilon^{0.75} > 4\epsilon$; or (iii) $\|\boldsymbol{A}(\tilde{t})\| \geqslant M_2\epsilon > 2\epsilon$.

Consider case (i),[8] from continuity there exists $t' \in \mathbb{R}$ such that $\|\boldsymbol{B}(t')\| = 4\epsilon$, and $t'' \in \mathbb{R}$ such that for any $t \in [t', t'']$, $4\epsilon \leqslant \|\boldsymbol{B}(t)\| \leqslant M_1\epsilon^{0.75}$. In such a case, we also have

$$|\dot{\gamma}| \leqslant 2kM\left(|\gamma| + 4\epsilon\sqrt{|\gamma|}\right) \leqslant 4kM|\gamma| \tag{E.71}$$

Integrating the inequality $|\dot{\gamma}| < 4kM|\gamma|$ by $t$ for $s \in [t', t'']$,

$$\int_{t'}^{s} \frac{1}{|\gamma|}|\dot{\gamma}|dt < \int_{t'}^{s} 4kM dt, \tag{E.72}$$

substituting integration variables and using $0 \leqslant t' \leqslant s \leqslant t'' \leqslant \ln\left(\frac{1}{\epsilon^{0.5}}\right)$,

$$\int_{\gamma(t')}^{\gamma(s)} \frac{1}{|\gamma|}d\gamma \leqslant 4kM\left(s - t'\right) < 4kM\left(\ln\left(\frac{1}{\epsilon^{0.5}}\right) - 0\right). \tag{E.73}$$

The above evaluates to,

$$\ln\left(\frac{|\gamma(s)|}{|\gamma(t')|}\right) < 4kM\ln\left(\frac{1}{\epsilon^{0.5}}\right) \tag{E.74}$$

which may be further manipulated to reach,

$$|\gamma(s)| < e^{4kM\ln\left(\frac{1}{\epsilon^{0.5}}\right)}|\gamma(t')| = (4\epsilon)^2 e^{4kM\ln\left(\frac{1}{\epsilon^{0.5}}\right)}. \tag{E.75}$$

where we have used $\|\boldsymbol{B}(t')\| = 4\epsilon$. The final bound on the norm of $\gamma(s)$ is therefore,

$$|\gamma(s)| < \frac{16\epsilon^2}{\epsilon^{0.5}}e^{4kM} = 16\epsilon^{1.5}e^{4kM}. \tag{E.76}$$

Denoting $M_1^2 = 16 \cdot e^{4kM}$, and taking the square root of the above,

$$\|\boldsymbol{B}(s)\| < M_1\epsilon^{0.75}. \tag{E.77}$$

We have shown that for all $0 \leqslant t \leqslant \bar{t} \leqslant \ln\left(\frac{1}{\epsilon^{0.5}}\right)$, there exists $M_1$ s.t $\|\boldsymbol{B}(t)\| < M_1\epsilon^{0.75}$ (the same proof applies for case (ii)).

Consider case (iii), we need to show that the bound over $\|\boldsymbol{A}\|_F$ applies for $t \in [0, \bar{t}]$. Notice that for a matrix, $\|\boldsymbol{A}\|_F^2 = Tr(\boldsymbol{A}^\top \boldsymbol{A})$, and

$$\frac{d}{dt}Tr(\boldsymbol{A}^\top \boldsymbol{A}) = Tr\left(\frac{d}{dt}(\boldsymbol{A}^\top \boldsymbol{A})\right) \tag{E.78}$$

$$= Tr\left(\dot{\boldsymbol{A}}^\top \boldsymbol{A}\right) + Tr\left(\boldsymbol{A}^\top \dot{\boldsymbol{A}}\right) \tag{E.79}$$

$$= 2Tr\left(\boldsymbol{A}^\top \dot{\boldsymbol{A}}\right), \tag{E.80}$$

where we have used the linearity of trace and its invariance to transpose. The derivative of $\boldsymbol{A}$ with respect to time (see Section E.1.4),

$$\dot{\boldsymbol{A}} = -\sum_{i=1}^{k-1} \nabla\ell_i \sum_{r=0}^{i-1} (\boldsymbol{A}^\top)^r \boldsymbol{C}^\top \boldsymbol{B}^\top (\boldsymbol{A}^\top)^{i-r-1}. \tag{E.81}$$

Multiplying it from the left by $\boldsymbol{A}^\top$ and then taking trace provides us with

$$Tr(\boldsymbol{A}^\top \dot{\boldsymbol{A}}) = -\sum_{i=1}^{k-1} \nabla\ell_i \sum_{r=0}^{i-1} Tr\left((\boldsymbol{A}^\top)^{r+1}\boldsymbol{C}^\top \boldsymbol{B}^\top (\boldsymbol{A}^\top)^{i-r-1}\right). \tag{E.82}$$

---

[8]The case of $\|\boldsymbol{C}(t)\| \geqslant M_1\epsilon^{0.75}$ is handled similarly.

Taking a transpose and then using the cyclic property of trace and, for each summand,

$$Tr\left((\boldsymbol{A}^\top)^{r+1}\boldsymbol{C}^\top\boldsymbol{B}^\top(\boldsymbol{A}^\top)^{i-r-1}\right) = Tr\left(\boldsymbol{B}^\top(\boldsymbol{A}^\top)^i\boldsymbol{C}^\top\right) \tag{E.83}$$

$$= Tr\left(\boldsymbol{C}\boldsymbol{A}^i\boldsymbol{B}\right) \tag{E.84}$$

$$= \boldsymbol{C}\boldsymbol{A}^i\boldsymbol{B}. \tag{E.85}$$

Equation E.82 evaluates to

$$Tr(\boldsymbol{A}^\top\dot{\boldsymbol{A}}) = -\sum_{i=1}^{k-1}\nabla\ell_i\sum_{r=0}^{i-1}\boldsymbol{C}\boldsymbol{A}^i\boldsymbol{B} = -\sum_{i=1}^{k-1}\nabla\ell_i\cdot i\cdot\boldsymbol{C}\boldsymbol{A}^i\boldsymbol{B}. \tag{E.86}$$

Bounding $Tr(\boldsymbol{A}^\top\dot{\boldsymbol{A}})$,

$$Tr(\boldsymbol{A}^\top\dot{\boldsymbol{A}}) \leqslant \sum_{i=1}^{k-1}|\nabla\ell_i|\cdot i\cdot|\boldsymbol{C}\boldsymbol{A}^i\boldsymbol{B}| \tag{E.87}$$

Using the Cauchy-Schwartz inequality and then plugging $M > |\nabla\ell_i|$ and the bounds found for $\|\boldsymbol{C}\|, \|\boldsymbol{B}\| < M_1\epsilon^{0.75}$ leads to

$$|\nabla\ell_i|\cdot i\cdot|\boldsymbol{C}\boldsymbol{A}^i\boldsymbol{B}| \leqslant M\cdot i\cdot\|\boldsymbol{C}\|\|\boldsymbol{A}\|_F^i\|\boldsymbol{B}\| \leqslant M\cdot i\cdot M_1^2\epsilon^{1.5}\|\boldsymbol{A}\|_F^i \tag{E.88}$$

Putting the bound of Equation E.88 into Equation E.87, results with,

$$Tr(\boldsymbol{A}^\top\dot{\boldsymbol{A}}) \leqslant \sum_{i=1}^{k-1}M\cdot i\cdot M_1^2\epsilon^{1.5}\|\boldsymbol{A}\|_F^i = M\cdot M_1^2\epsilon^{1.5}\sum_{i=1}^{k-1}i\cdot\|\boldsymbol{A}\|_F^i \tag{E.89}$$

Noting that $\|\boldsymbol{A}\|_F < 1$ and denoting $\tilde{M}_2 = \frac{k^2}{2}M\cdot M_1^2$ leads to

$$\sum_{i=1}^{k-1}i\|\boldsymbol{A}\|_F^i < \|\boldsymbol{A}\|_F\frac{k(k-1)}{2} < \frac{k^2}{2}\|\boldsymbol{A}\|_F \tag{E.90}$$

Therefore, we get that

$$Tr(\boldsymbol{A}^\top\dot{\boldsymbol{A}}) < \tilde{M}_2\epsilon^{1.5}\|\boldsymbol{A}\|_F. \tag{E.91}$$

Notice that

$$\|\boldsymbol{A}\|_F\cdot\frac{d}{dt}(\|\boldsymbol{A}\|_F) = 0.5\frac{d}{dt}(\|\boldsymbol{A}\|_F^2) = Tr(\boldsymbol{A}^\top\dot{\boldsymbol{A}}) \tag{E.92}$$

$$\Rightarrow \|\boldsymbol{A}\|_F\cdot\frac{d}{dt}(\|\boldsymbol{A}\|_F) < \tilde{M}_2\epsilon^{1.5}\|\boldsymbol{A}\|_F. \tag{E.93}$$

$$\Rightarrow \frac{d}{dt}(\|\boldsymbol{A}\|_F) < \tilde{M}_2\epsilon^{1.5} \tag{E.94}$$

Therefore, for any $0 \leqslant s \leqslant \bar{t}$,

$$\|\boldsymbol{A}(s)\|_F - \|\boldsymbol{A}(0)\|_F = \int_0^s\frac{d}{dt}(\|\boldsymbol{A}(t)\|_F)dt < \tilde{M}_2\epsilon^{1.5}s, \tag{E.95}$$

$$\Rightarrow \|\boldsymbol{A}(s)\|_F < \tilde{M}_2\epsilon^{1.5}\ln\left(\frac{1}{\epsilon^{0.5}}\right) + \|\boldsymbol{A}(0)\|_F. \tag{E.96}$$

We make use of the fact that $\forall x > 0, \ln(x) < x$ to bound,

$$\tilde{M}_2\epsilon^{1.5}\ln\left(\frac{1}{\epsilon^{0.5}}\right) \leqslant \tilde{M}_2\epsilon^{1.5}\epsilon^{-0.5} = \tilde{M}_2\epsilon. \tag{E.97}$$

From our assumption on initialization, $\|\boldsymbol{A}(0)\|_F < 2\epsilon$. Putting back together,

$$\|\boldsymbol{A}(s)\|_F < \tilde{M}_2\epsilon + 2\epsilon. \tag{E.98}$$

Taking $M_2 = \tilde{M}_2 + 2$, we have for all $0 \leqslant t \leqslant \bar{t}$ that

$$\|\boldsymbol{A}(t)\|_F < M_2\epsilon, \tag{E.99}$$

concluding the proof.

$$\square$$

### E.4.1 Bounding the differential equations

***Proof of Lemma E.14.*** Denote

$$\boldsymbol{Y} \equiv \boldsymbol{C}^\top - \boldsymbol{B}, \qquad \boldsymbol{W} \equiv \boldsymbol{C}^\top + \boldsymbol{B}. \tag{E.100}$$

Recall that

$$\frac{\partial \mathcal{L}}{\partial \boldsymbol{B}} = \sum_{i=0}^{k-1} \nabla \ell_i (\boldsymbol{A}^\top)^i \boldsymbol{C}^\top, \qquad \frac{\partial \mathcal{L}}{\partial \boldsymbol{C}} = \boldsymbol{B}^\top \sum_{i=0}^{k-1} \nabla \ell_i (\boldsymbol{A}^\top)^i \tag{E.101}$$

We can write the change in $\boldsymbol{Y}$,

$$\dot{\boldsymbol{Y}} = \dot{\boldsymbol{C}}^\top - \dot{\boldsymbol{B}} = -\sum_{i=0}^{k-1} \nabla \ell_i \boldsymbol{A}^i \boldsymbol{B} + \sum_{i=0}^{k-1} \nabla \ell_i (\boldsymbol{A}^\top)^i \boldsymbol{C}^\top \tag{E.102}$$

$$= \sum_{i=0}^{k-1} \nabla \ell_i \left( (\boldsymbol{A}^\top)^i \boldsymbol{C}^\top - \boldsymbol{A}^i \boldsymbol{B} \right)$$

Denote $(\boldsymbol{A}^i)_S = \frac{\boldsymbol{A}^i + (\boldsymbol{A}^\top)^i}{2}$ and $(\boldsymbol{A}^i)_{\bar{S}} = \frac{\boldsymbol{A}^i - (\boldsymbol{A}^\top)^i}{2}$ the symmetric and anti-symmetric parts of $\boldsymbol{A}^i$. We can now write

$$(\boldsymbol{A}^\top)^i \boldsymbol{C}^\top - \boldsymbol{A}^i \boldsymbol{B} = \left[ (\boldsymbol{A}^i)_S - (\boldsymbol{A}^i)_{\bar{S}} \right] \boldsymbol{C}^\top - \left[ (\boldsymbol{A}^i)_S + (\boldsymbol{A}^i)_{\bar{S}} \right] \boldsymbol{B} \tag{E.103}$$

$$= (\boldsymbol{A}^i)_S (\boldsymbol{C}^\top - \boldsymbol{B}) - (\boldsymbol{A}^i)_{\bar{S}} (\boldsymbol{C}^\top + \boldsymbol{B})$$

$$= (\boldsymbol{A}^i)_S \boldsymbol{Y} - (\boldsymbol{A}^i)_{\bar{S}} \boldsymbol{W}$$

Note also that for $i = 0$, we have $\boldsymbol{A}^0 = I$ and its anti-symmetric part is the zero matrix, writing $i = 0$ separately and assigning equation E.103 into equation E.102,

$$\dot{\boldsymbol{Y}} = \nabla \ell_0 \boldsymbol{Y} + \sum_{i=1}^{k-1} \nabla \ell_i \left( (\boldsymbol{A}^i)_S \boldsymbol{Y} - (\boldsymbol{A}^i)_{\bar{S}} \boldsymbol{W} \right) \tag{E.104}$$

Let us look at $\frac{d}{dt} (\boldsymbol{Y}^\top \boldsymbol{Y}) = \dot{\boldsymbol{Y}}^\top \boldsymbol{Y} + \boldsymbol{Y}^\top \dot{\boldsymbol{Y}} = 2 \boldsymbol{Y}^\top \dot{\boldsymbol{Y}}$.

Multiplying equation E.104 from the left with $\boldsymbol{Y}^\top$ evaluates to

$$\boldsymbol{Y}^\top \dot{\boldsymbol{Y}} = \nabla \ell_0 \boldsymbol{Y}^\top \boldsymbol{Y} + \sum_{i=1}^{k-1} \nabla \ell_i \left( \boldsymbol{Y}^\top (\boldsymbol{A}^i)_S \boldsymbol{Y} - \boldsymbol{Y}^\top (\boldsymbol{A}^i)_{\bar{S}} \boldsymbol{W} \right) \tag{E.105}$$

We now turn to bound the terms in the sum.

$$\boldsymbol{Y}^\top \dot{\boldsymbol{Y}} \leqslant \nabla \ell_0 \boldsymbol{Y}^\top \boldsymbol{Y} + \sum_{i=1}^{k-1} |\nabla \ell_i| \left( |\boldsymbol{Y}^\top (\boldsymbol{A}^i)_S \boldsymbol{Y}| + |\boldsymbol{Y}^\top (\boldsymbol{A}^i)_{\bar{S}} \boldsymbol{W}| \right) \tag{E.106}$$

We can bound each term using Cauchy–Schwarz. We first need to bound $\|\boldsymbol{Y}\|$ and $\|\boldsymbol{W}\|$, which are trivially bounded by

$$\|\boldsymbol{Y}\|, \|\boldsymbol{W}\| \leqslant \|\boldsymbol{C}\| + \|\boldsymbol{B}\| \leqslant 2 M_1 \epsilon^{0.75} \tag{E.107}$$

As for the symmetric and anti-symmetric parts of $\boldsymbol{A}$,

$$\|(\boldsymbol{A}^i)_S\|_F = \left\| \frac{\boldsymbol{A}^i + (\boldsymbol{A}^i)^\top}{2} \right\|_F \leqslant \frac{1}{2} \left( \|\boldsymbol{A}^i\|_F + \|(\boldsymbol{A}^i)^\top\|_F \right) = \|\boldsymbol{A}^i\|_F \leqslant \|\boldsymbol{A}\|_F^i, \tag{E.108}$$

where the last inequality follows again from Cauchy–Schwarz (the same considerations apply for $(\boldsymbol{A}^i)_{\bar{S}}$).

From Cauchy–Schwarz we can bound $|\boldsymbol{Y}^\top (\boldsymbol{A}^i)_{\bar{S}} \boldsymbol{W}| \leqslant \|\boldsymbol{Y}^\top\| \|(\boldsymbol{A}^i)_{\bar{S}}\|_F \|\boldsymbol{W}\| \leqslant \|\boldsymbol{Y}\| \|\boldsymbol{A}\|_F^i \|\boldsymbol{W}\|$, denote $M_3 = max\{M_1, M_2\}$ and derive,

$$|\boldsymbol{Y}^\top (\boldsymbol{A}^i)_{\bar{S}} \boldsymbol{W}| \leqslant 2 M_1 \epsilon^{0.75} (M_2 \epsilon)^i 2 M_1 \epsilon^{0.75} < 4 M_3^3 \epsilon^{1.5+i}, \tag{E.109}$$

which is maximized when $i = 1$. We again bound: $M = \max_i(|w_i|) + M_1^2\epsilon^{1.5} > |\nabla\ell_i|$. We can bound the terms in equation E.106 by

$$|\nabla\ell_i|\left(|\boldsymbol{Y}^\top(\boldsymbol{A}^i)_S\boldsymbol{Y}| + |\boldsymbol{Y}^\top(\boldsymbol{A}^i)_{\bar{S}}\boldsymbol{W}|\right) \leqslant 8 \cdot M \cdot M_3^3\epsilon^{2.5} \qquad (E.110)$$

Plugging back into equation E.106:

$$\boldsymbol{Y}^\top\dot{\boldsymbol{Y}} \leqslant \nabla\ell_0\boldsymbol{Y}^\top\boldsymbol{Y} + \sum_{i=1}^{k-1} 8 \cdot M \cdot M_3^3\epsilon^{2.5} \qquad (E.111)$$

We can also bound $\nabla\ell_0 = (\boldsymbol{CB} - w_0) \leqslant -w_0 + |\boldsymbol{CB}| \leqslant -w_0 + \|\boldsymbol{C}\|\|\boldsymbol{B}\| \leqslant -w_0 + M_1^2\epsilon^{1.5}$. Note also that we multiply by $\boldsymbol{Y}^\top\boldsymbol{Y}$ so we can bound

$$\nabla\ell_0\boldsymbol{Y}^\top\boldsymbol{Y} \leqslant -w_0\boldsymbol{Y}^\top\boldsymbol{Y} + \underbrace{M_1^2\epsilon^{1.5}(M_21\epsilon^{0.75})^2}_{=4M_1^4\epsilon^3} \qquad (E.112)$$

putting back together, we get

$$\boldsymbol{Y}^\top\dot{\boldsymbol{Y}} \leqslant -w_0\boldsymbol{Y}^\top\boldsymbol{Y} + 4M_1^4\epsilon^3 + (k-1)(M_1+1)8 \cdot M \cdot M_3^3\epsilon^{2.5} \qquad (E.113)$$

In particular, there exists $M_4$ such that

$$\boldsymbol{Y}^\top\dot{\boldsymbol{Y}} \leqslant -w_0\boldsymbol{Y}^\top\boldsymbol{Y} + M_4\epsilon^{2.5} \qquad (E.114)$$

Recall that we were interested in bounding $\frac{d}{dt}(\boldsymbol{Y}^\top\boldsymbol{Y}) = \dot{\boldsymbol{Y}}^\top\boldsymbol{Y} + \boldsymbol{Y}^\top\dot{\boldsymbol{Y}} = 2\boldsymbol{Y}^\top\dot{\boldsymbol{Y}}$,

$$\frac{d}{dt}(\boldsymbol{Y}^\top\boldsymbol{Y}) \leqslant -2w_0\boldsymbol{Y}^\top\boldsymbol{Y} + M_4\epsilon^{2.5} \qquad (E.115)$$

Denoting $z(t) \equiv \boldsymbol{Y}(t)^\top\boldsymbol{Y}(t)$ and $x(t) \equiv \boldsymbol{W}(t)^\top\boldsymbol{W}(t)$, and using Lemma E.15, we have the desired bounds

$$z(t) \leqslant \frac{1}{2w_0}\left[(33w_0d\epsilon^2)e^{-2w_0t} + M_4\epsilon^{2.5}\right] \qquad (E.116)$$

In particular, we can write

$$z(t) \leqslant c_1\epsilon^2 e^{-2w_0t} + c_2\epsilon^{2.5} \qquad (E.117)$$

and

$$x(t) \geqslant c_3\epsilon^2 e^{2w_0t} - c_4\epsilon^{2.5} \qquad (E.118)$$

where $c_i$'s are positive constants.

Note that the derivation of $x(t)$ is exactly the same as $z(t)$ with opposite signs and bounding from below instead.

$\square$

### E.4.2 INTEGRAL BOUND OF DIFFERENTIAL EQUATIONS

**Lemma E.15.** *Assume* $\dot{z} < -2w_0z + M_4\epsilon^{2.5} < 0, \dot{x} > 2w_0z - M_4\epsilon^{2.5} > 0$ *, where* $w_0 > 0$*. Then, under the assumptions of Lemma E.10:*

$$z(t_1) < \frac{1}{2w_0}\left(\exp(-2w_0t_1) \cdot (75w_0d\epsilon^2) + M_4\epsilon^{2.5}\right) \qquad (E.119)$$

$$x(t_2) > \frac{1}{2w_0}\left(\exp(2w_0t_2) \cdot (\frac{w_0\epsilon^2}{25d}) + M_4\epsilon^{2.5}\right) \qquad (E.120)$$

***Proof of Lemma E.15.*** Assume $\dot{z} < -2w_0z + M_4\epsilon^{2.5}$ , where $w_0 > 0, 2w_0z > M_4\epsilon^{2.5}$. Similarly, assume $\dot{x} > 2w_0x - M_4\epsilon^{2.5}$.

Then:

$$\frac{\dot{z}}{2w_0z - M_4\epsilon^{2.5}} < -1 \qquad (E.121)$$

$$\frac{\dot{x}}{2w_0 x - M_4 \epsilon^{2.5}} > 1 \tag{E.122}$$

Integrating both sides by dt, and using integration by substitution, we get:

$$-t_1 = \int_0^{t_1} -1 > \int_0^{t_1} \frac{1}{2w_0 z - M_4 \epsilon^{2.5}} \frac{dz}{dt} dt = \int_{z(0)}^{z(t_1)} \frac{1}{2w_0 z - M_4 \epsilon^{2.5}} dz \tag{E.123}$$

$$t_2 = \int_0^{t_2} 1 < \int_0^{t_2} \frac{1}{2w_0 x - M_4 \epsilon^{2.5}} \frac{dx}{dt} dt = \int_{x(0)}^{x(t_2)} \frac{1}{2w_0 x - M_4 \epsilon^{2.5}} dx \tag{E.124}$$

We note:

$$\int_{z(0)}^{z(t_1)} \frac{1}{2w_0 z - M_4 \epsilon^{2.5}} dz = \frac{1}{2w_0} [\ln(2w_0 z(t_1) - M_4 \epsilon^{2.5}) - \ln(2w_0 z(0) - M_4 \epsilon^{2.5})] \tag{E.125}$$

$$\Rightarrow \int_{z(0)}^{z(t_1)} \frac{1}{2w_0 z - M_4 \epsilon^{2.5}} dz = \frac{1}{2w_0} \left[ \ln\left( \frac{2w_0 z(t_1) - M_4 \epsilon^{2.5}}{2w_0 z(0) - M_4 \epsilon^{2.5}} \right) \right] \tag{E.126}$$

$$\int_{x(0)}^{x(t_2)} \frac{1}{2w_0 x - M_4 \epsilon^{2.5}} dx = \frac{1}{2w_0} \left[ \ln\left( \frac{2w_0 x(t_2) - M_4 \epsilon^{2.5}}{2w_0 x(0) - M_4 \epsilon^{2.5}} \right) \right] \tag{E.127}$$

Combining equations, we have:

$$\frac{1}{2w_0} \left[ \ln\left( \frac{2w_0 z(t_1) - M_4 \epsilon^{2.5}}{2w_0 z(0) - M_4 \epsilon^{2.5}} \right) \right] < -t_1 \tag{E.128}$$

$$\Rightarrow \ln\left( \frac{2w_0 z(t_1) - M_4 \epsilon^{2.5}}{2w_0 z(0) - M_4 \epsilon^{2.5}} \right) < -2w_0 t_1 \tag{E.129}$$

$$\Rightarrow \frac{2w_0 z(t_1) - M_4 \epsilon^{2.5}}{2w_0 z(0) - M_4 \epsilon^{2.5}} < \exp(-2w_0 t_1) \tag{E.130}$$

$$\Rightarrow 2w_0 z(t_1) - M_4 \epsilon^{2.5} < \exp(-2w_0 t_1) \cdot (2w_0 z(0) - M_4 \epsilon^{2.5}) \tag{E.131}$$

$$\Rightarrow z(t_1) < \frac{1}{2w_0} \left[ \exp(-2w_0 t_1) \cdot (2w_0 z(0) - M_4 \epsilon^{2.5}) + M_4 \epsilon^{2.5} \right] \tag{E.132}$$

$$x(t_2) > \frac{1}{2w_0} \left[ \exp(2w_0 t_2) \cdot (2w_0 x(0) - M_4 \epsilon^{2.5}) + M_4 \epsilon^{2.5} \right] \tag{E.133}$$

Note that $z(0) = \mathbf{Y}^\top(0)\mathbf{Y}(0)$. By linearity of sum of variances, $\mathbf{Y}(0)$'s entries are distributed according to $\mathcal{N}(0, \sqrt{2} \cdot \epsilon)$, by Lemma E.10:

$$\frac{\sqrt{2}}{2} \epsilon < \|\mathbf{Y}(0)\| < \frac{3\sqrt{2}}{2} \epsilon \tag{E.134}$$

From Cauchy-Schwartz, $z(0) < 3\epsilon^2$ with high probability. $\mathbf{W}$ is distributed as $\mathbf{Y}$, therefore, $x(0) > \frac{1}{2}\epsilon^2$. Assuming $M_4 \epsilon^{2.5} < \frac{w_0 \epsilon^2}{2}$, we have:

$$z(t_1) < \frac{1}{2w_0} \left( \exp(-2w_0 t_1) \cdot (6w_0 \epsilon^2) + M_4 \epsilon^{2.5} \right) \tag{E.135}$$

$$x(t_2) > \frac{1}{2w_0} \left( \exp(2w_0 t_2) \cdot (w_0 \epsilon^2) + M_4 \epsilon^{2.5} \right) \tag{E.136}$$

Concluding the proof. $\square$

