# OpenReview forum: "Learning Low Dimensional State Spaces with Overparameterized Recurrent Neural Nets"
_ICLR.cc/2023/Conference — ICLR 2023 poster_

### Official Review · Reviewer_3zQj · 2022-10-22

**Confidence:** 3
**Correctness:** 4
**Technical Novelty And Significance:** 3
**Empirical Novelty And Significance:** 1
**Recommendation:** 6

**Clarity, Quality, Novelty And Reproducibility:**

**Clarity.**
The manuscript is written very well and is a smooth read overall. Both the introduced concepts as well as the mathematical proofs are clearly presented and easy to follow. Occasionally, some things could be improved, some concrete thoughts are in the review down below.

**Quality.**
The proofs are clearly writte, well supported and follow a nice line of thoughts. The experiments are well designed, but results are only presented for a relatively small number of trials and problem sizes.

**Novelty.**
The central conceptional contribution is the connection of the population loss of linear RNNs to a moment matching objective, which I appreciate a lot. That being said, the idea is very fundamental and I originally thought it could have been studied in the literature on parameter identification for linear dynamical systems. That being said, I am not an expert in this area and a quick search did not reveal a work giving this intepretation directly.

**Reproducibility.**
The experiments are well described in the appendix, however, no code is provided.


**Strength And Weaknesses:**

Strengths:
* The manuscript provides a nice connection of the population loss for parameter estimation in linear RNNs as a moment matching objective. This could be a useful tool for future theoretical works.
* The theoretical analysis is strong, well supported and clearly laid out.

Weaknesses:
* *Convergence of training:* A study of convergence properties of training dynamics is not provided. The main contribution of the manuscript is an extrapolation result for balanced students and teachers rather than a result for solutions obtained by gradient descent training. When comparing this to the study of overparametrized networks in the context of supervised learning the manuscript can not explain why overparametrized students are superior. One argument could be improved convergence for these models.
* *Experiments:* The experimental evaluation of the results could be strengthened in several aspects. Currently, the results shown and discussed in Section 5 are only for a single student size. I think the experiments can easily be strengthened by including more student sizes, more random trials and more sequence lengths. More importantly, I think the relation to the theoretical findings is not discussed in great detail, i.e., whether the transition to perfect extrapolation occurs at the predicted sequence length is not discussed sufficiently. Finally, it could be interesting to add experiments with fixed sequence length and increasing student size since in practice one will have data of a fixed length and will need to set the student size.

**Summary Of The Paper:**

The manuscript studies extrapolation properties when learning the parameters of a linear dynamical system when predicting outputs on time horizons longer than present in the training data in a student teacher setup.
For balanced parameters the population loss over a time horizon $k$ is re-interpreted as the summed quadratic distance between the first $k$-th moments of two random variables associated to the students and teachers parameters.
This allows the application of results from moment matching theory, which guarantee that all moments of the random variables match, when the first $k$ moments do, where $k$ needs to be sufficiently large in dependency of the teachers state space dimension. Noting that balancedness is preserved under gradient flow (GF) implies that zero loss solutions found by GF starting from balanced parameters exhibits perfect interpolation. An approximate version of the result on perfect extrapolation is also provided.
The theoretical findings are supported by experiments on different models are conducted in order to show the phase transition between perfect and non perfect extrapolation.

**Summary Of The Review:**

The manuscript  provides a valuable contribution in studying the extrapolation properties of linear RNNs through a connection of the population loss of linear RNNs to a moment matching problem.
My main criticism is presented under **weaknesses**, which concerns the absence of a study of the convergence properties of GF dynamics as well as the realm of the experiments. Where my thoughts regarding additional experiments are outlined above, I want to share a few thoughts on the main message of the paper here:

I believe that the main theorem of the paper is Lemma 7 and Theorem 5 is a corollary of this. Hence, I wonder, whether it might be better to present the results as extrapolation results for balanced students. The same applies to Theorem 8.

Some other things that caught my eye:
* In Theorem 5, Lemma 7 etc: Do you require $d>k$? Probably this is included since for these regimes Proposition 3 implies that the problem of extrapolation is hard. Nevertheless, putting this as an assumption can convey the message that the result is not true for $d\le k$.
* In the experiments in Subsectino 5.1: Your empirical results indicate perfect interpolation is reached somewhere between $k = 16$ and $k = 20$, where your theoretical analysis suggest perfect extrapolation for $k>2\hat d = 10$. Is my reading of this correct? If not, what is the correct interpretation of your empirical results? If so, what is your interpretation of the results?
* The second to last paragraph in the conclusion section should be in the discussion of related works or introduction. No new things should be discussed in the conclusion.
* No actual content should be in footnotes or brackets, e.g., footnote 2 should be in the main text. Also in „Figure 1(a) reports the extrapolation error (quantified by the $\ell^\infty$ distance between
the impulse response of the learned student and that of the teacher) as a function of $k$.“ I would simply delete the brackets.

---

> ### Author Response · Authors · 2022-11-16
> **Response to Reviewer 3zQj (1/2)**
>
> Thank you for the feedback and suggestions! We address all points below.
>
> ### **Convergence of training:**
>
> - **“A study of convergence properties of training dynamics is not provided”.** We agree that studying convergence (i.e. establishing that GD minimizes the loss) is of interest.  Please note however that in the context of implicit generalization and implicit extrapolation, assuming convergence without proof (as we do in this work) is extremely common (cf. Woodworth et al., 2020; Azulay et al., 2021; Brutzkus & Globerson 2017; Razin & Cohen 2020; Razin et al., 2021; Razin et al., 2022; Gunasekar et al., 2017; Li et al., 2021).  Nevertheless, our paper does include a partial analysis of the dynamics of the system. See for example Proposition 9 that shows GD will arrive at a symmetric solution at some time point.
>
> - **“Contribution of the manuscript is an extrapolation result for balanced students and teachers rather than a result for solutions obtained by gradient descent”.** While we agree that our main technical contribution can be seen as Lemma 7, its relevance to practical deep learning critically relies on specific properties of GD — namely the fact that it preserves balancedness (Lemma 6), as well as its tendency to approach balancedness when initialized near zero (Proposition 9).
>
> - **“Manuscript can not explain why overparametrized students are superior”.** Our theory shows that, in the analyzed setting, a student perfectly extrapolates if it is sufficiently overparameterized with respect to the teacher and the length of training sequences.  We thus establish a benefit for students whose overparameterization surpasses a certain threshold.  We agree with your statement in the sense that we do not show an advantage for students overparameterized beyond this threshold.  It is possible that an additional advantage of over-parameterization is that it makes it easier to optimize the training loss, but as stated above, this aspect is beyond the scope of our work.
>
> ---
>
> ### **Experiments:**
> - **“The experiments can easily be strengthened by including more student sizes, more random trials and more sequence lengths”** As stated in the text, in addition to the students for which results are reported, larger students were also evaluated, and the results they yielded were essentially the same.  We will include in our experimental figures more student sizes, random trials and sequence lengths, as you suggest.  Thank you!
>
> - **“Relation to the theoretical findings is not discussed in great detail, i.e., whether the transition to perfect extrapolation occurs at the predicted sequence length is not discussed sufficiently”.** We mention several times throughout the text that our empirical findings demonstrate a phase transition in extrapolation, which in the linear case is around $k = 2 \hat{d}$, in compliance with our theory (see Figure 1).  You are correct that we do not discuss the fact that the phase transition takes place gradually (rather than sharply at $k = 2 \hat{d}$).  We associate this fact to numerical phenomena, in particular the loss not reaching the exact global minimum.  A short discussion along this line was added to the manuscript.  Thank you for raising this point!
>
> ---
>
> **“The idea is very fundamental and I originally thought it could have been studied in the literature on parameter identification for linear dynamical systems”.** To the best of our knowledge, the idea of interpreting RNNs as random variables has not been proposed earlier.  We stress that this interpretation relies on the balancedness property preserved by GD. We are happy that you appreciate this idea, and agree that it may prove to be a useful tool for future theoretical works.

---

> > ### Author Response · Authors · 2022-11-16
> > **Response to Reviewer 3zQj (2/2)**
> >
> > **Additional points:**
> > - **“I believe that the main theorem of the paper is Lemma 7 and Theorem 5 is a corollary of this. Hence, I wonder, whether it might be better to present the results as extrapolation results for balanced students. The same applies to Theorem 8”.** While Theorems 5 and 8 are indeed corollaries of Lemma 7, we chose to frame them as theorems since (in our view) from a deep learning perspective they are the most important results.
> > - **“In Theorem 5, Lemma 7 etc: Do you require d>k?”.** We indeed require $d > k$, and this is explicitly specified in the statements of Theorems 5 and Lemma 7.  Please let us know if any further clarification on this matter is needed.
> > - **“In the experiments in Subsectinon 5.1: Your empirical results indicate perfect interpolation is reached somewhere between k=16 and k=20, where your theoretical analysis suggest perfect extrapolation for $k>2\hat{d}=10$. Is my reading of this correct?”** See our response to your general comment about experiments.
> > - **"The second to last paragraph in the conclusion section should be in the discussion of related works or introduction. No new things should be discussed in the conclusion".** We have moved said paragraph to the related work section.  Thank you!.
> > - **No actual content should be in footnotes or brackets, e.g., footnote 2 should be in the main text. Also in Figure 1(a) reports the extrapolation error (quantified by the ℓ∞ distance between the impulse response of the learned student and that of the teacher) as a function of k. I would simply delete the brackets".** - We have implemented these changes.  Thank you!
> >
> > **References**
> > - Moritz Hardt, Tengyu Ma, and Benjamin Recht. Gradient descent learns linear dynamical systems. arXiv preprint arXiv:1609.05191, 2016.
> > - Zeyuan Allen-Zhu, Yuanzhi Li, and Zhao Song. On the convergence rate of training recurrent neural networks. Advances in neural information processing systems (NeurIPS), 2019.
> > - Blake Woodworth, Suriya Gunasekar, Jason D Lee, Edward Moroshko, Pedro Savarese, Itay Golan, Daniel Soudry, and Nathan Srebro. Kernel and rich regimes in overparametrized models. In Conference on Learning Theory (COLT), 2020.
> > - Shahar Azulay, Edward Moroshko, Mor Shpigel Nacson, Blake E Woodworth, Nathan Srebro, Amir Globerson, and Daniel Soudry. On the implicit bias of initialization shape: Beyond infinitesimal mirror descent. In International Conference on Machine Learning (ICML), 2021.
> > - Alon Brutzkus and Amir Globerson. Globally Optimal Gradient Descent for a ConvNet with Gaussian Inputs. In International Conference on Machine Learning (ICML), 2017.
> > - Noam Razin and Nadav Cohen. Implicit regularization in deep learning may not be explainable by norms. In Advances in Neural Information Processing Systems (NeurIPS), 2020.
> > - Noam Razin, Asaf Maman, and Nadav Cohen. Implicit regularization in tensor factorization. International Conference on Machine Learning (ICML), 2021.
> > - Noam Razin, Asaf Maman, and Nadav Cohen. Implicit regularization in hierarchical tensor factorization and deep convolutional neural networks. International Conference on Machine Learning (ICML), 2022.
> > - Suriya Gunasekar, Blake E Woodworth, Srinadh Bhojanapalli, Behnam Neyshabur, and Nati Srebro. Implicit regularization in matrix factorization. Advances in Neural Information Processing Systems (NeurIPS), 2017.
> > - Zhiyuan Li, Yuping Luo, and Kaifeng Lyu. Towards resolving the implicit bias of gradient descent for matrix factorization: Greedy low-rank learning. In International Conference on Learning Representations (ICLR), 2021.

---

> > > ### Comment · Reviewer_3zQj · 2022-12-11
> > > **Thanks for your reply**
> > >
> > > Dear authors,
> > >
> > > thanks a lot for your reply!
> > >
> > > I will keep my weak accept score. The main reason is that I think the experimental side can still be improved considerably. Most importantly, I think the following things would -- as mentioned in my initial review -- make the manuscript stronger:
> > >
> > > * Incorporating more student sizes / trials (I guess you would be adding this for a camera ready version, but in the revised version it is not present in Figure 1). As suggested, also an experiment with fixed sequence length and increasing student sizes would be insightful.
> > > * Still, you state that in Figure 1 on the left / subsection 5.1 'when $k > 2 \hat d$ extrapolation error is low'. I can still not agree with this, since for $k = 11, 12 > 2\hat d=10$ extrpolation error as shown in Figure 1 left is still over $30\%$ and in the order same of magnitude compared to $k< 2\hat d$.
> > >
> > > I wish the authors all the best,

---

> > > > ### Author Response · Authors · 2022-12-13
> > > > **Additional remarks**
> > > >
> > > > Thank you for the response!
> > > > - As stated in the manuscript (first paragraph of Section 5.2), we have experimented with different student sizes, and observed essentially the same results as those shown in Section 5.  Per your suggestion, we will add a figure with fixed sequence length and varying student sizes to the camera-ready version.
> > > > - Regarding the phase transition in extrapolation, while our theory indeed predicts perfect extrapolation for $k > 2\hat{d}$, it relies on training achieving a perfect fit (zero loss).  This is not the case in practice, hence the phase transition occurs gradually.  We will add a note along this line to the camera-ready version. Thank you!

---

### Official Review · Reviewer_n2fU · 2022-10-22

**Confidence:** 3
**Correctness:** 4
**Technical Novelty And Significance:** 3
**Empirical Novelty And Significance:** 2
**Recommendation:** 6

**Clarity, Quality, Novelty And Reproducibility:**

The paper is very clear and seems reproducible, the results are novel to the best of my knowledge.

**Strength And Weaknesses:**

**Strengths:**

The paper is very well written, and relatively easy to follow in spite of the highly technical nature of the results presented.

The main result is remarkably simple to state, and yet rather nontrivial to prove.

I appreciate a lot the addition of numerical results to supplement, strengthen and extend the theoretical findings. I think nowadays this should be a mandatory practice, while too often it is overlooked or ignored by theoretical studies.


**Weaknesses:**

The results presented are very technical and limited in their scope. In particular, although probably the assumptions used here are less strong than those used in previous theoretical works on the same topic, the main theorems proven are still based on quite strong and unrealistic assumptions such as the linearity of the investigated RNNs and the complete independence of the terms in the sequences.

However, I am aware of the fact that deriving rigorous theoretical results for state of the art (nonlinear) neural networks is a remarkably difficult task, and, moreover, the numerical tests seem to show that the key findings of the work could extend beyond the restricted theoretical setting analysed.




**Summary Of The Paper:**

This work provides some theoretical results for the phenomenon of RNNs “extrapolation” i.e., the ability of RNNs to provide correct predictions on test sequences longer than the sequences available at training time.

The theory is built with a simple teacher-student setup with the following properties:

- Teacher and student are represented as linear RNNs.

- The teacher and student RNNs have states spaces of dimension $\hat{d}$ and $d$  respectively.

- The length of the sequences available at training time is $k$.

- The student RNN is in the overparametrised regime ($d > k$).

- No correlation exists between the elements of the input sequences.

- The student RNN is initialised in a “balanced” condition, implying that the linear operators defining the RNN are connected in a specific way that also helps in the calculations.

Under the assumptions and the notation specified above, the authors prove the following main result:

If $k > 2d $ (the training sequences are greater than 2 times the teacher state space dimension) and if a gradient flow loss minimisation reaches zero quadratic loss, then the parameters obtained necessarily generalise.

The authors then enhance this result by relaxing the zero loss condition provided that the teacher is “stable”, and finally prove that the balancedness condition is approximately implied by a near zero initialisation of the RNNs weights, thus justifying one of their assumptions.

After the above theorems are stated/proved, several numerical results are presented.

These consist in set of curves showing the extrapolation error as a function of the sequence length ($k$) for fixed teacher dimension ($d$) and for several teacher-student settings, specifically:

(1) the theoretical settings used to prove the theorems

(2) a simplified settings where certain assumptions from (1) are relaxed (such as the balancedness of the teacher)

(3) a nonlinear RNN setting where essentially none of the simplifying assumptions used hold exactly

For all settings (and remarkably also for the nonlinear setting (3)), the curves resemble a phase transition where the generalisation error decreases sharply around a specific "critical" value of $k$. The critical $k$ value found numerically is around $2d$, in agreement with the theory, trivially for setting (1) and notably also for setting (2) where some theory assumptions have been relaxed. For setting (3) instead, the critical $k$ does not fall around $2d$ but around $4d$.

**Summary Of The Review:**

I appreciate the elegance of the results obtained, the clarity of the writing, the robustness of the results. I also appreciate the presence of several numerical experiments to support and extend the key findings to broader and more realistic settings.

I highlight the somewhat obvious limitations of this work in terms of its strong and unrealistic assumptions, but I recognise the difficulty of theoretical analysis in more realistic settings, and further recognise that the numerical experiments provided seem to suggest that the key findings of this work could extend beyond the restricted setting where they have been proven.

I believe this is a good publication for ICLR. It is not a breakthrough publication, but it is a technically very solid paper which can have good impact in specific research communities.

---

> ### Author Response · Authors · 2022-11-16
> **Response to Reviewer n2fU**
>
> Thank you very much for the detailed and positive review!
>
> While (as you suggest) our theory is more general than those of prior works, we agree that its scope is still limited, in the sense that it treats simplified settings.  We appreciate you recognizing the difficulty of theoretically analyzing realistic RNNs, as well as the fact that our experiments suggest that key findings of our theory extend beyond the specific settings to which it applies.  We would like to note that in the context of feed-forward NNs, there have been cases where theoretical analyses for simplified settings involving linear activations were later extended to cover more general settings with non-linear models (see for example Lyu & Li (2019), Ji & Telgarsky (2020) and Razin et al. (2022)).  Encouraged by these extensions, we believe our work may serve as a stepping stone towards analyzing non-linear RNNs, and are currently exploring this proposition.
>
>
>
> Given your positive stance towards our paper, we would greatly appreciate it if you would consider raising your score.  Thank you!
>
> **References**
> - Kaifeng Lyu and Jian Li. Gradient descent maximizes the margin of homogeneous neural networks. In International Conference on Learning Representations (ICLR), 2019.
> - Ziwei Ji and Matus Telgarsky. Gradient descent aligns the layers of deep linear networks. In International Conference on Learning Representations (ICLR), 2018.
> - Noam Razin, Asaf Maman, and Nadav Cohen. Implicit Regularization in Hierarchical Tensor Factorization and Deep Convolutional Neural Networks. In International Conference on Machine Learning (ICML), 2022.

---

### Official Review · Reviewer_C8pz · 2022-10-24

**Confidence:** 2
**Correctness:** 4
**Technical Novelty And Significance:** 2
**Empirical Novelty And Significance:** Not applicable
**Recommendation:** 5

**Clarity, Quality, Novelty And Reproducibility:**

The paper is clear, although the math feels a bit too convoluted at times. I am not familiar enough with this particular area to judge the novelty of this approach.

**Strength And Weaknesses:**

The mathematical arguments seem solid. However, there results only apply to linear RNNs in the teacher-student framework. Even there, conditions on the teacher are necessary to proof the authors statements. I feel the studied setup is too restrictive and it would need to tackle at least non-linear RNNs or more complex learning scenarios. Also, the insights don't lead to any new algorithm/model.

**Summary Of The Paper:**

The authors study from a theoretical point of view the extrapolation capabilities of linear RNNs in the teacher-student framework. They show that under certain conditions, students with 0 error may not extrapolate to longer time horizons than the ones they were trained on. They also show, that under the right parameterization of the teacher, if Gradient Flow converges to a zero loss in the training, the student will be able to extrapolate perfectly. They also prove that under some more conditions, the student $\epsilon$-extrapolates.

**Summary Of The Review:**

It's a well-written paper that does a thorough analysis of linear RNNs in the teacher-student framework. However, I feel the scope of the paper is too narrow to justify acceptance at ICLR. If the authors would either study a more general RNN setup or build some new training algorithm based on the insights, this could be a strong publication.

---

> ### Author Response · Authors · 2022-11-16
> **Response to Reviewer C8pz**
>
> Thank you for the feedback. We address points raised below.
>
> 1. **“Studied setup is too restrictive and it would need to tackle at least non-linear RNNs or more complex learning scenarios”, “scope of the paper is too narrow to justify acceptance at ICLR”.** Analyzing simplified settings towards understanding more complex ones is extremely common in the theory of deep learning.  In particular, it is very common to study linear NNs under different assumptions on their architecture, initialization, and data.  Indeed, numerous works along this line have recently appeared in top-tier machine learning publication venues (Arora et al., 2018; Shamir 2019; Arora et al., 2019a), in particular ICLR (Ji & Telgarsky 2018; Brutzkus et al., 2018; Arora et al., 2019b; Lyu & Li 2020; Li et al., 2020).   Note that while the vast majority of existing analyses for linear NNs treat feed-forward architectures, we study the recurrent setting, which arguably is far more challenging.  We empirically demonstrated our theoretical conclusions with non-linear RNNs, and believe it may be possible to formally analyze such models through extension of our theory, similarly to how theories for linear feed-forward NNs were later extended to account for non-linear feed-forward NNs (see for example Lyu & Li (2019), Ji & Telgarsky (2020) and Razin et al. (2022)).
> 2. **“If the authors would … build some new training algorithm based on the insights, this could be a strong publication”.** While we do not propose a new training algorithm, we believe our results deliver insights that may assist practitioners. For example, our results suggest that when learning RNNs, it suffices for the length of training sequences to be on the order of the “intrinsic dimension” of the data generating distribution.  Put differently, if an RNN with a hidden dimension n has sufficient representational capacity for fitting the data distribution, then there is no need for the length of training sequences to go beyond $O(n)$, irrespective of how large the actual learned RNN is.  This observation may reduce the computational and storage costs of both data collection and training. Additionally, we believe that understanding why currently used learning algorithms (e.g., gradient descent) work is a key open problem in deep learning nowadays, and in the case of learning RNNs, very little is understood. Yet another takeaway from our work is the importance of small initialization in RNNs.
> 3. **“The math feels a bit too convoluted at times”.** We are happy to address specific comments you may have concerning our mathematical derivation.  Please let us know if there are such.
>
> **References**
> - Sanjeev Arora, Nadav Cohen, and Elad Hazan. On the optimization of deep networks: Implicit acceleration by overparameterization. In International Conference on Machine Learning, (ICML), 2018.
> - Alon Brutzkus, Amir Globerson, Eran Malach, and Shai Shalev-Shwartz. SGD learns overparameterized networks that provably generalize on linearly separable data. In International Conference on Learning Representations (ICLR), 2018.
> - Ohad Shamir. Exponential convergence time of gradient descent for one-dimensional deep linear neural networks. In Conference of Learning Theory (COLT), 2019.
> - Sanjeev Arora, Nadav Cohen, Wei Hu, and Yuping Luo. Implicit regularization in deep matrix factorization. In Advances in Neural Information Processing Systems (NeurIPS), 2019a.
> - Kaifeng Lyu and Jian Li. Gradient descent maximizes the margin of homogeneous neural networks. In International Conference on Learning Representations (ICLR), 2019.
> - Zhiyuan Li, Yuping Luo, and Kaifeng Lyu. Towards resolving the implicit bias of gradient descent for matrix factorization: Greedy low-rank learning. In International Conference on Learning Representations (ICLR), 2020.
> - Sanjeev Arora, Nadav Cohen, Noah Golowich, and Wei Hu. A convergence analysis of gradient descent for deep linear neural networks. International Conference on Learning Representations (ICLR), 2019b.
> - Ziwei Ji and Matus Telgarsky. Gradient descent aligns the layers of deep linear networks. In International Conference on Learning Representations (ICLR), 2018.
> - Chulhee Yun, Shankar Krishnan, and Hossein Mobahi. A unifying view on implicit bias in training linear neural networks. In International Conference on Learning Representations (ICLR), 2020.

---

### Official Review · Reviewer_8yoF · 2022-10-25

**Confidence:** 3
**Clarity, Quality, Novelty And Reproducibility:** This paper is novel and clear. The ov…
**Correctness:** 4
**Technical Novelty And Significance:** 4
**Empirical Novelty And Significance:** Not applicable
**Recommendation:** 6

**Strength And Weaknesses:**

Strength: The topic is interesting. The result seems convincing. The presentation is relatively clear.
Weaknesses: Some places in the setup need more justification. See comments below.

It is reasonable to investigate SISO linear systems as an initial step to understanding the performance of actual RNNs. Most of the parts are clearly stated and well-written. I have some details questions as follows.

1. In Eq. (3.1), why does the transition matrix $A$ need to be symmetric (i.e., $A^T=A$)?

2. In Eq. (3.2) and Eq. (3.3), the loss uses the output only at the last time step. I wonder why not use the output of all time steps to calculate the loss.

3. The authors mention some existing studies of implicit extrapolation in linear RNNs suggesting that GD is biased toward solutions with short-term memory. Does this contradict the result in this paper that low dimensional state space does not coincide the short-term memory? If so, how to explain such a contraction?

**Summary Of The Paper:**

This paper studies a single-input single-output (SISO) linear system that has a similar sequential input-output structure as the recurrent neural network (RNN). The ground-truth labels are generated by a teacher which is assumed to be a similar linear system. The authors model gradient descent (GD) with small step size by gradient flow (GF), and prove that the convergence of GF to a zero loss solution leads the student to extrapolate (i.e., learn the ground truth completely even beyond the training horizon) when the training sequence length is greater than two times of the teacher's state space dimension, regardless of how large the student state space dimension is.

**Summary Of The Review:**

The paper is well-written and provides some novel results, yet some places need more clarification.

---

> ### Author Response · Authors · 2022-11-16
> **Response to Reviewer 8yoF**
>
> Thank you for the positive and thoughtful feedback!
> Points raised are addressed below.
> 1. **In Eq. (3.1), why does the transition matrix A need to be symmetric?** The transition matrix A needs to be symmetric in order for our proof techniques to be applicable (for example, interpreting the impulse response of the RNN as moments of a random variable requires A to be orthogonally diagonalizable with real eigenvalues).  As stated in the text, restriction to symmetric transition matrices is customary in both theory (Hazan et al., 2018) and practice (Gupta et al., 2022), and represents a generalization of the *canonical modal form*.  We believe it is possible to extend our results to the case where A is non-symmetric, and view this as an interesting direction for future work.
>
> 2. **In Eq. (3.2) and Eq. (3.3), the loss uses the output only at the last time step. I wonder why not use the output of all time steps to calculate the loss.** The loss is based on the output at the last time step merely for simplicity of presentation.  Our results can easily be adapted to account for a loss based on the output across all time steps.  We have modified the text to clarify this point.  Thank you for raising it!
>
> 3. **The authors mention some existing studies of implicit extrapolation in linear RNNs suggesting that GD is biased toward solutions with short-term memory. Does this contradict the result in this paper that low dimensional state space does not coincide the short-term memory? If so, how to explain such a contraction?** Past works suggesting that GD over linear RNNs is biased towards short-term memory, namely Cohen-Karlik et al. (2022) and Emami et al. (2021), analyzed settings more restrictive than ours (in particular, Cohen-Karlik et al. considered the case of a memoryless teacher, and Emami et al. studied a scenario where the teacher is stable and its impulse response decays exponentially fast).  In these settings, implicit extrapolation via learning low dimensional state spaces indeed leads to solutions with short-term memory, so there is NO contradiction between our work and Cohen-Karlik et al. (2022); Emami et al. (2021).  However, in a more general setting where the assumptions of a memoryless teacher and exponentially decaying impulse response are not met, learning low dimensional state spaces still leads to extrapolation, while the learned solution does NOT have short-term memory.  We have clarified this point in the revised manuscript; thank you for bringing it up!
>
> Given your positive stance towards the paper, and in the hopes that we have properly addressed your questions (please let us know if this is not the case), we would greatly appreciate it if you would consider raising your score.  Thank you!
>
> **References**
> - Elad Hazan, Holden Lee, Karan Singh, Cyril Zhang, and Yi Zhang. Spectral filtering for general linear dynamical systems. Advances in Neural Information Processing Systems (NeurIPS), 2018.
> - Ankit Gupta, Albert Gu, and Jonathan Berant. Diagonal state spaces are as effective as structured state spaces. arXiv preprint arXiv:2203.14343, 2022.
> - Edo Cohen-Karlik, Avichai Ben David, Nadav Cohen, and Amir Globerson. On the implicit bias of gradient descent for temporal extrapolation. International Conference on Artificial Intelligence and Statistics (AISTATS), 2022.
> - Melikasadat Emami, Mojtaba Sahraee-Ardakan, Parthe Pandit, Sundeep Rangan, and Alyson K. Fletcher. Implicit bias of linear rnns. In International Conference on Machine Learning, (ICML), 2021.

---

### Decision · Program_Chairs · 2023-01-20

**Decision:**

Accept: poster

**Justification For Why Not Higher Score:**

The paper has a restrictive setup which does not merit a higher score.

**Justification For Why Not Lower Score:**

The paper is an interesting step in the right direction. Clearly borderline with some flaws but I think the text of the reviews are a bit lower than the written statements.

**Metareview: Summary, Strengths And Weaknesses:**

In this paper, the authors analyze the extrapolation properties of gradient descent (GD) when applied to overparameterized linear recurrent neural networks (RNNs). They provide theoretical evidence for learning low-dimensional state spaces that can model long-term memory, in contrast to recent arguments suggesting an implicit bias towards short-term memory. Their results rely on a dynamical characterization of GD and tools developed in the context of the moment problem from statistics. Experiments corroborate their theory, demonstrating extrapolation via learning low-dimensional state spaces with both linear and non-linear RNNs.

The reviewers thought the problem is interesting, paper is well written for the most part and that investigating SISO linear systems is a good step towards understanding RNNs. Some reviewers thought the scope of the paper is too narrow and focusing only on the linear case is too restrictive and that the authors make "strong and unrealistic assumptions". I read the paper carefully myself. I agree that the assumptions of the paper are restrictive. However, I think the paper is a step in the right direction with an interesting analysis. Therefore, I recommend acceptance.

**Note From Pc:**

if the above contains the word "oral" or "spotlight" please see: "oral" presentation means -> notable-top-5% and "spotlight" means -> notable-top-25%. As stated in our emails, we are disassociating presentation type from AC recommendations

**Summary Of Ac-Reviewer Meeting:**

Due to timezone difference I was not able to hold an in person meeting. I did however, solicit discussions and read the paper carefully myself. The reviewers are sticking to their original score. Based on their feedback I think the paper is clearly borderline but IMO the benefits outweigh the flaws. I think is an interesting direction and merits publication.